# Measurement report: Oxidation potential of water-soluble aerosol components in the southern and northern of Beijing

Wei Yuan[1], Ru-Jin Huang[1], Chao Luo[2], Lu Yang[1], Wenjuan Cao[1], Jie Guo[1], Huinan Yang[2]

[1]State Key Laboratory of Loess and Quaternary Geology, Center for Excellence in Quaternary Science and Global Change, Institute of Earth Environment, Chinese Academy of Sciences, Xi'an 710061, China.

[2]School of Energy and Power Engineering, University of Shanghai for Science and Technology, Shanghai 200093, China

Correspondence: Ru-Jin Huang (rujin.huang@ieecas.cn) and Huinan Yang (yanghuinan@usst.edu.cn)

**Abstract**

Water-soluble components have significant contribution to the oxidative potential (OP) of atmospheric fine particles (PM$_{2.5}$), while our understanding of water-soluble PM$_{2.5}$ OP and its sources, as well as its relationship with water-soluble components, is still limited. In this study, the water-soluble OP levels in wintertime PM$_{2.5}$ in the south and north of Beijing, representing the difference in sources, were measured with dithiothreitol (DTT) assay. The volume normalized DTT (DTT$_v$) in the north (3.5 ± 1.2 nmol min$^{-1}$ m$^{-3}$) was comparable to that in the south (3.9 ± 0.9 nmol min$^{-1}$ m$^{-3}$), while the mass normalized DTT (DTT$_m$) in the north (65 ± 28 pmol min$^{-1}$ µg$^{-3}$) was almost twice that in the south (36 ± 14 pmol min$^{-1}$ µg$^{-3}$). In both the south and north of Beijing, DTT$_v$ was better correlated with soluble elements instead of total elements. In the north, soluble elements (mainly Mn, Co, Ni, Zn, As, Cd and Pb) and water-soluble organic compounds, especially light-absorbing compounds (also known as brown carbon), had positive correlations with DTT$_v$. However, in the south, the

DTT$_v$ was mainly related to soluble As, Fe and Pb. The sources of DTT$_v$ were further
resolved using the positive matrix factorization (PMF) model. Traffic-related
emissions (39%) and biomass burning (25%) were the main sources of DTT$_v$ in the
south, and traffic-related emissions (> 50%) contributed the most of DTT$_v$ in the north.
Our results indicate that vehicle emission was the important contributor to OP in
Beijing ambient PM$_{2.5}$ and suggest that more study is needed to understand the
intrinsic relationship between OP and light absorbing organic compounds.

**1 Introduction**
Atmospheric fine particulate matter (PM$_{2.5}$) pollution is one of the major global
environmental issues, affecting air quality, climate and human health (Huang et al.,
2014; Burnett et al., 2018; An et al., 2019; Zheng et al., 2020). The exposure to PM$_{2.5}$
was estimated to be responsible for 8.9 million deaths worldwide in 2015, of which
28% occurred in China (Burnett et al., 2018). Numerous studies have shown that
oxidative stress is one possible mechanisms underlying the adverse effects of PM$_{2.5}$
on human health (Chowdhury et al., 2019; Lelieveld et al., 2021; Yu et al., 2022b;
Guascito et al., 2023). When entering the human body, PM$_{2.5}$ can induce the
production of excessive reactive oxygen species (ROS) (e.g., H$_2$O$_2$, ·OH and ·O$_2^-$),
leading to cellular redox imbalance and generating oxidative stress effects. The ability
of PM$_{2.5}$ to cause oxidative stress is defined as oxidative potential (OP).
The methods to determine the OP of PM$_{2.5}$ include cellular and acellular assays,
and acellular methods are more widely used than cellular methods (Charrier and
Anastasio, 2012; Xiong et al., 2017; Calas et al., 2018; Bates et al., 2019; Wang et al.,
2020b; Campbell et al., 2021; Oh et al., 2023). Among acellular methods, the
dithiothreitol (DTT) assay is extensively applied to determine the OP of ambient
particles (Charrier and Anastasio, 2012; Xiong et al., 2017; Liu et al., 2018; Wang et
al., 2020b; Puthussery et al., 2022; Wu et al., 2022a). DTT is a surrogate of cellular
reductants, and the consumption rate of DTT was used to assess the OP of PM$_{2.5}$.
Previous studies have shown that organic matters (e.g., water-soluble organic species

and PAHs) and some transition metals (e.g., Mn and Cu) are the important contributors to DTT consumption of $PM_{2.5}$ (Charrier and Anastasio, 2012; Verma et al., 2015; Bates et al., 2019; Wu et al., 2022a; Wu et al., 2022b). For example, Charrier and Anastasio (2012) measured the OP of $PM_{2.5}$ in San Joaquin Valley, California and reported that about 80% of DTT consumption was contributed by transition metals. Verma et al. (2015) measured the OP of water-soluble $PM_{2.5}$ in the southeastern United States and reported that about 60% of DTT activity was contributed by water-soluble organics. The mixtures of metals and organics may produce synergistic or antagonistic effects, such as $\cdot O_2^-$ produced from oxidation of DTT by quinones is more efficiently transformed to $\cdot OH$ in the presence of Fe, while the DTT consumption and $\cdot OH$ generation of quinones are reduced in the presence of Cu (Xiong et al., 2017; Yu et al., 2018; Bates et al., 2019).

A number of studies have investigated the OP of water-soluble components in $PM_{2.5}$, which show that the average water-soluble OP values in urban areas ranged from 0.1 to 10 nmol $min^{-1}$ $m^{-3}$ (Fang et al., 2016; Liu et al., 2018; Chen et al., 2019; Wu et al., 2022a; Yu et al., 2022a; Xing et al., 2023). Due to the complexity in chemical composition and sources of $PM_{2.5}$ that determine the OP levels, the sources of OP are also diverse (Verma et al., 2015; Bates et al., 2019; Tuet et al., 2019; Yu et al., 2019; Cao et al., 2021). Several studies have investigated the emission sources and ambient samples to identify the sources of OP (Tuet et al., 2019; Yu et al., 2019; Wang et al., 2020b; Cao et al., 2021), which include both primary and secondary sources. For example, Cao et al. (2021) measured the water-soluble OP of $PM_{2.5}$ samples from six biomass and five coal burning emissions in China, with average values of 4.5-7.4 and 0.5-2.1 pmol $min^{-1}$ $\mu g^{-1}$, respectively. Tong et al. (2018) investigated the OP of secondary organic aerosols (SOA) from oxidation of naphthalene, isoprene and $\beta$-pinene with $\cdot OH$ or $O_3$, which were $104 \pm 7.6$, $48 \pm 7.9$ and $36 \pm 3.1$ pmol $min^{-1}$ $\mu g^{-1}$, respectively. Verma et al. (2014) identified the sources of water-soluble OP of $PM_{2.5}$ in Atlanta, United States from June 2012 to September 2013 with positive matrix factorization (PMF) and chemical mass balance (CMB) methods, of which biomass

burning was the largest contributor. Wang et al. (2020b) quantified the sources of
water-soluble OP of $PM_{2.5}$ in Xi'an, China in 2017 using PMF and multiple linear
regression (MLR) methods, with significant contributions from secondary sulfates,
vehicle emissions and coal combustion. Some studies have also measured the OP of
particles with different particle sizes, and reported that smaller size fractions typically
have higher ROS activity compared to large PM size fractions (Saffari et al., 2014;
Shafer et al., 2016; Besis et al., 2023). For example, Besis et al. (2023) measured the
OP of water-soluble fraction of size segregated PM (< 0.49, 0.49-0.95, 0.95-1.5, 1.5-
3.0, 3.0-7.2 and > 7.2 μm) collected during the cold and warm periods at an urban site
in Thessaloniki, northern Greece, and the results showed that the total DTT activity of
the PM < 3 μm size fraction were higher (2-5 times) than that of PM > 3 μm size
fraction in both warm and cold periods. Despite these efforts, comparative studies on
the differences in pollution levels and sources of $PM_{2.5}$ OP in different districts are
still limited.
In this study, the DTT activity of water-soluble matter in $PM_{2.5}$ samples collected
simultaneously in the southern and northern of Beijing in January 2018 were
measured. The concentration and light absorption of water-soluble organic carbon
(WSOC), as well as the concentrations of 14 trace elements and 7 light-absorbing
nitroaromatic compounds (NACs) were quantified. The sources of DTT activity were
then identified with PMF model. The results acquired in this study provide a
comparison of $PM_{2.5}$ OP in different districts of Beijing and its connection with
organic compounds, trace elements and sources, which could be helpful for further
study of the regional differences in the effects of $PM_{2.5}$ on human health.

**2 Materials and methods**
**2.1 Sampling**
Ambient 24 h integrated $PM_{2.5}$ filter samples were collected from January 1 to 31,
2018 simultaneously in the south (the Dingfuzhuang village (DFZ), Daxing district;
39.61°N, 116.28°E) and north (the National Center for Nanoscience and Technology

(NCNT), Haidian district; 39.99°N, 116.32°E) of Beijing (Figure S1). The distance between the two sampling sites is about 42 km. The south site is surrounded by agricultural, industrial, and transportation areas, and the north site is surrounded by residential, transportation and commercial areas. $PM_{2.5}$ samples were collected on pre-baked (780 °C, 3 h) quartz-fiber filters (20.3 × 25.4 cm; Whatman, QM-A, Clifton, NJ, USA) using high-volume $PM_{2.5}$ samplers (1.13 $m^{-3}$ $min^{-1}$; Tisch, Cleveland, OH, USA) which were placed on the roof of buildings at heights of about 5 m (south) and 20 m (north) above the ground. 31 samples were collected at each site. After collection, the samples were wrapped in baked aluminum foils and stored in a freezer (−20 °C) until further analysis.

**2.2 Chemical analysis**

The mass of $PM_{2.5}$ on the filter was measured by a digital microbalance with a precision of 0.1 mg (LA130S-F, Sartorius, Germany) after 24-h equilibration at a constant temperature (20-23 °C) and humidity (35-45%) chamber. Each filter was weighted at least two times, and the deviations for blank and sampled filters among the repetitions were less than 5 and 10 μg, respectively. The $PM_{2.5}$ mass concentration was calculated by dividing the weight difference before and after sampling by the volume of sampled air.

For WSOC analysis, one punch (1.5 $cm^2$ for concentration analysis and 0.526 $cm^2$ for light absorption measurement) of filter was taken from each sample and extracted ultrasonically with ultrapure water (> 18.2 MΩ cm) for 30 min. After, the extracts were filtered with a 0.45 μm PVDF pore syring filter to remove insoluble substances. Finally, the concentration of WSOC was measured with a total organic carbon-total nitrogen analyzer (TOC-L, Shimadzu, Japan; (Ho et al., 2015)) and the light absorption of WSOC was measured by an UV-Vis spectrophotometer (300-700 nm; Ocean Optics, USA) equipped with a liquid waveguide capillary cell (LWCC-3100, World Precision Instruments, Sarasota, FL, USA; (Yuan et al., 2020)). The absorption coefficient (Abs) of WSOC was calculated according to formula S1 in the Supporting Information (SI).

145  The total concentration and soluble fraction concentration of 14 trace elements

146 (i.e., Ti, V, Cr, Mn, Fe, Co, Ni, Cu, Zn, As, Sr, Cd, Ba, and Pb) were quantified by an

147 inductively coupled plasma mass spectrometer (ICP-MS, 7700x, Agilent Technologies,

148 USA), and the details are shown in the SI. For soluble fraction concentration analysis,

149 a punch of filter (47 mm diameter) was extracted with ultrapure water and then

150 centrifuged from residues. For total concentration analysis, another 47 mm diameter

151 filter of the same sample was used and digested with 10 mL $HNO_3$ and 1 mL HF at

152 180 °C for 12 h. The extracts were then heated and concentrated to ~ 0.1 mL, and

153 diluted to 5 mL with 2% $HNO_3$. Afterwards, the diluents were filtered with a 0.22 μm

154 PTFE pore syring filter and stored in a freezer (−4 °C) until further ICP-MS analysis.

155  The concentrations of organic markers (including levoglucosan, mannosan,

156 galactosan, hopanes (including 17α(H)-22,29,30-trisnorhopane, 17α(H),21β(H)-30-

157 norhopane, 17β(H),21α(H)-30-norhopane, 17β(H),21α(H)-hopane, 17β(H),21α(H)-

158 hopane and 17β(H),21α(H)-hopane), picene, phthalic acid, isophthalic acid and

159 terephthalic acid) and light-absorbing NACs (including 4-nitrophenol (4NP), 2-

160 methyl-4-nitrophenol (2M4NP), 3-methyl-4-nitrophenol (3M4NP), 4-nitrocatechol

161 (4NC), 3-methyl-5-nitrocatechol (3M5NC), 4-methyl-5-nitrocatechol (4M5NC) and

162 4-nitro-1-naphthol (4N1N)) were determined by a gas chromatograph–mass

163 spectrometer (GC-MS; Agilent Technologies, Santa Clara, CA, USA) following the

164 method described elsewhere (Wang et al., 2020a), and more details about the analysis

165 can be found in SI. All of the results reported in this study were corrected for blanks.

166 **2.3 Oxidative potential**

167  The DTT assay was applied to determine the oxidative potential of water-soluble

168 components in $PM_{2.5}$ according to the method by Gao et al. (2017). In brief, a quarter

169 of a 47 mm filter was ultrasonically extracted with 5 mL ultrapure water for 30 min

170 and then filtered with a 0.45 μm PVDF pore syring filter to remove insoluble

171 substances. Several studies have shown that ultrasonic treatment of samples can lead

172 to an increase in their OP values (Miljevic et al., 2014; Jiang et al., 2019), however,

173 there was also a study showed that the difference in OP values of water-soluble $PM_{2.5}$

measured by DTT assay was small for samples extracted by ultrasonic and shaking
(Gao et al., 2017). Consistent with the extraction methods for organic markers and
trace elements, ultrasonic method was used to extract samples for DTT analysis.
Afterwards, 0.5 mL of the extract was mixed with 1 mL of potassium phosphate
buffer (pH = 7.4) and 0.5 mL of 2 mM DTT in a brown vial, and then placed in a
water bath at 37 °C. Then, 20 μL of this mixture was taken at designated time
intervals (2, 7, 13, 20, and 28 min) and mixed with 1 mL trichloroacetic acid (TCA;
1% w/v) in another brown vial to terminate the reaction. Then, 0.5 mL of 5,5'-
dithiobis-(2-nitrobenzoic acid) (DTNB; 2.5 μM) and 2 mL of tris buffer (pH = 8.9)
were added to form 2-nitro-5-thiobenzonic acid (TNB) which has light absorption at
412 nm. Finally, the absorption of TNB was measured by a LWCC-UV-Vis. The DTT
consumption rate was quantified by the remaining DTT concentration at different
reaction times. Daily solution blanks and filter blanks were analyzed in parallel with
samples to evaluate the consistency of the system performance. Besides, for every 10
samples, one sample was chosen to be measured three times to check the
reproducibility, and the relative standard deviation was lower than 5%. Ambient
samples were corrected for filter blank. The DTT activities were normalized by the
volume of sampled air ($DTT_v$, nmol min$^{-1}$ m$^{-3}$) and the mass concentration of $PM_{2.5}$
($DTT_m$, pmol min$^{-1}$ μg$^{-1}$).

Considering that for samples containing a significant amount of substances

whose DTT response is non-linear with $PM_{2.5}$ concentration (e.g., Cu, Mn), the $DTT_m$
value depends on the concentration of $PM_{2.5}$ added to the reaction solution (Charrier
et al., 2016). The response of $DTT_m$ to $PM_{2.5}$ concentration added to the reaction
solution was analyzed using sample containing high concentrations of soluble Cu and
Mn (Figure S2). When the $PM_{2.5}$ concentration added to the reaction solution is less
than 150 μg mL$^{-1}$, the $DTT_m$ response is greatly affected by the difference in added
$PM_{2.5}$ concentration; however, when the $PM_{2.5}$ concentration added to the reaction
solution is greater than 150 μg mL$^{-1}$, the $DTT_m$ response is less affected by the
difference in $PM_{2.5}$ concentration (< 12%). In this study, the concentration of $PM_{2.5}$
added to the reaction solution of most samples from the two sites was greater than 150
$\mu g\ mL^{-1}$ (ranged from 79 to 749 $\mu g\ mL^{-1}$, with an average of 409 ± 164 and 207 ± 95
$\mu g\ mL^{-1}$ in the south and north, respectively), therefore, the difference in $PM_{2.5}$
concentration added to the reaction solution of different samples should had a
relatively small impact on the difference in $DTT_m$ values of different samples. This
study did not consider the impact of metal precipitation in phosphate matrix on the
measured DTT values, as there is not a straightforward method to correct the artifact
caused by this phenomenon (Yalamanchili et al., 2023).
**2.4 Source apportionment**
The sources of DTT activities were identified and quantified using PMF model
implemented by the multilinear engine (ME-2; (Paatero, 1997)) following the method
described in our previous studies (Huang et al., 2014; Yuan et al., 2020). For each site,
31 samples (a total of 62 samples) and 23 species were input into PMF model. The
number of samples is higher than the number of species. The input data include
species concentration (including $DTT_v$, 14 trace elements and 8 organic markers) and
uncertainties. The species-specific uncertainties were calculated following Liu et al.
(2017). For a clear separation of sources profiles, the contribution of corresponding
markers was set to 0 in the sources unrelated to the markers (see Table S1). More
details are described in SI (PMF analysis).

**3 Results and discussion**
**3.1 DTT activity and concentrations of water-soluble $PM_{2.5}$ components**
Figure 1 shows the daily variation of DTT activity, light absorption of WSOC at
wavelength 365 nm ($Abs_{365}$), together with the concentrations of $PM_{2.5}$, WSOC,
NACs and total elements in the south and north of Beijing. Their average values are
shown in Table S2. Generally, the average values of $PM_{2.5}$, WSOC, $Abs_{365}$, NACs and
total elements were higher in the south than in the north. Specifically, the
concentrations of $PM_{2.5}$ and WSOC in the south (122 ± 49 $\mu g\ m^{-3}$ and 8.1 ± 5.0 $\mu gC$
$m^{-3}$, respectively) were both about two times higher than that in the north (62 ± 28 $\mu g$

m$^{-3}$ and 4.0 ± 2.0 μgC m$^{-3}$, respectively), indicating that the proportion of WSOC in PM$_{2.5}$ was similar in the south and north. However, the Abs$_{365}$ in the south was about three times that in the north, indicating that the chemical composition of WSOC was different between the south and north. Previous studies have reported that NACs are the main water-soluble light-absorbing organic compounds (also known as brown carbon, BrC) of PM$_{2.5}$ (Lin et al., 2017; Huang et al., 2020; Li et al., 2020). For the 7 NACs quantified in this study, the total concentration of nitrophenols (4NP, 2M4NP and 3M4NP), nitrocatechols (4NC, 3M5NC and 4M5NC), and 4N1N in the south (108 ± 73 ng m$^{-3}$, 118 ± 91 ng m$^{-3}$ and 12 ± 8.2 ng m$^{-3}$, respectively) was about three, five and four times, respectively, those in the north (35 ± 22 ng m$^{-3}$, 24 ± 30 ng m$^{-3}$ and 3.1 ± 3.0 ng m$^{-3}$, respectively). These results indicate that the sources and emission strength of water-soluble organic compounds were different in the south and north of Beijing, suggesting the different contribution of water-soluble organic compounds to DTT activity. The concentration trends of total trace elements were also different between the south and north of Beijing, with Fe > Zn > Ti > Mn > Cu > Ba > Pb > Sr > Cr > As > V > Ni > Cd > Co in the south, and Fe > Ti > Zn > Ba > Mn > Pb > Cu > Cr > Sr > As > Ni > V > Cd > Co in the north. It should be noted that although the contents of PM$_{2.5}$, WSOC and total elements measured in this study were higher in the south than in the north, the average DTT$_v$ value in the south (3.9 ± 0.9 nmol min$^{-1}$ m$^{-3}$) was comparable to that in the north (3.5 ± 1.2 nmol min$^{-1}$ m$^{-3}$), meanwhile, the average DTT$_m$ value was much higher (1.8 times) in the north (65 ± 28 pmol min$^{-1}$ μg$^{-1}$) than in the south (36 ± 14 pmol min$^{-1}$ μg$^{-1}$). Ahmad et al. (2021) also reported that the concentrations of PM$_{2.5}$, WSOC, and most elements in Lahore, Pakistan, were higher than those in Peshawar, Pakistan, while the DTT$_v$ values of the two sites were similar, and the DTT$_m$ value in Peshawar was higher than that in Lahore. The lower DTT$_m$ in the south than in the north may be due to the increased PM$_{2.5}$ in the south containing more substances with no or little contribution to DTT activity, and indicates that the intrinsic OP of water-soluble components of PM$_{2.5}$ was higher in the north than in the south. The similar DTT$_v$ values in the south and north

indicate that the exposure-relevant OP of water-soluble components of $PM_{2.5}$ was
comparable in the two sites, and the water-soluble $DTT_v$ was not consistent with the
content of water-soluble substances. Due to the complex chemical composition of
$PM_{2.5}$, there may also be antagonistic and synergistic effects, contributing to the
inconsistent relationship between DTT activity and compounds content (Xiong et al.,
2017; Lionette et al., 2021).

Figure 2 shows the comparison of water-soluble $PM_{2.5}$ DTT activity measured in

this study with those measured in other regions of Asia during similar periods. It can
be seen that the $DTT_v$ values measured in Beijing in this study were lower than that in
Jinzhou, Tianjin, Yantai, and Shanghai in China, Lahore and Peshawar in Pakistan,
and Delhi in India (Liu et al., 2018; Ahmad et al., 2021; Puthussery et al., 2022; Wu et
al., 2022a), higher than that in Xi'an, Nanjing, Hangzhou, Guangzhou, and Shenzhen
in China (Wang et al., 2019; Wang et al., 2020b; Ma et al., 2021; Yu et al., 2022c;
Xing et al., 2023), and comparable with that in Ningbo, China (Chen et al., 2022).
Different from $DTT_v$, the $DTT_m$ value measured in NCNT in Beijing was similar with
that in Jinzhou, Tianjin, Yantai, Shanghai and Ningbo in China (Liu et al., 2018; Chen
et al., 2022; Wu et al., 2022a), and higher than that in other regions. The differences in
water-soluble DTT activity of $PM_{2.5}$ in different regions can be explained by the
differences in chemical composition, sources and atmospheric formation processes
(Tong et al., 2017; Wong et al., 2019; Daellenbach et al., 2020; Wang et al., 2020b;
Cao et al., 2021). For example, Cao et al. (2021) reported the water-soluble DTT
activity of $PM_{2.5}$ from biomass and coal burning emissions in China, and the average
value of biomass burning (4.5-7.4 $pmol\ min^{-1}\ \mu g^{-1}$) was much higher than that of coal
burning (0.5-2.1 $pmol\ min^{-1}\ \mu g^{-1}$). Tuet et al. (2017) measured the water-soluble DTT
activity of SOA generated under different precursors and reaction conditions, with
SOA from naphthalene photooxidation under $RO_2$ + NO-dominant dry reaction
conditions had the highest DTT activity.
**3.2 Correlation between DTT activity and water-soluble $PM_{2.5}$ components**

Figure 3 shows the correlations of $DTT_v$ with $PM_{2.5}$, WSOC and $Abs_{365}$ in the

south and north of Beijing. It can be seen that the correlation coefficient between $DTT_v$ and $PM_{2.5}$ was moderate in both the south (r = 0.42) and north (r = 0.45), indicating that the OP of particles cannot be evaluated solely by the total $PM_{2.5}$ concentration. The correlations between $DTT_v$ with WSOC and $Abs_{365}$ were strong in the north (r of 0.69 and 0.70, respectively), while relatively weak in the south (r of 0.41 and 0.40, respectively). The high correlations between $DTT_v$ with WSOC and $Abs_{365}$ in the north of Beijing qualitatively agree with previous studies in Xi'an, China and Atlanta, United States (Verma et al., 2012; Chen et al., 2019), and suggest that water-soluble organic matter, especially BrC, has a significant contribution to DTT consumption in the north. Light-absorbing BrC typically has conjugated electrons, making it more likely to transport electrons for catalytic reactions, thereby contributing to DTT activity (Chen et al., 2019; Wu et al., 2022). Further, in the north, the $DTT_v$ was closely related to the concentrations of NACs (r of 0.57 to 0.79) (Figure S3), suggesting that NACs may be important contributors to DTT consumption. Feng et al. (2022) reported the positive correlations between NACs and biomarkers in saliva and urine (interleukin-6 and 8-hydrox-2′-deoxyguanosine). Zhang et al. (2023) also reported that NACs are major proinflammatory components in organic aerosols, contributing about 24% of the interleukin-8 response of all compounds detected by Fourier transform ion cyclotron resonance mass spectrometry (FT-ICR-MS) in electrospray ionization negative mode (ESI-). Certainly, it may also be other substances related to NACs that contribute to the DTT activity, including those not detected in this study, driving the good correlation between NACs and $DTT_v$ in the north of Beijing, which is worth studying in the future.

The correlation coefficients between $DTT_v$ and 14 trace elements are shown in Figure 4. Generally, the correlations between $DTT_v$ and soluble elements were higher than that between $DTT_v$ and total elements in both the south and north of Beijing. For soluble elements, in the south, the $DTT_v$ showed positive correlations with Mn, Fe, Cr, Co, As and Pb (r > 0.5), while in the north, it exhibited strong positive correlations with Mn, Co, Ni, Zn, As, Cd and Pb (r > 0.7), indicating the different sources of $DTT_v$

in the south and north of Beijing. It is worth noting that the concentrations of all soluble elements were higher in the south than in the north (Figure S4), while the correlation between $DTT_v$ and most soluble elements was lower in the south than in the north (Figure 4). The high correlations between $DTT_v$ and soluble elements in the north of Beijing suggests that soluble elements also had a significant contribution to DTT consumption. The low correlations between $DTT_v$ and soluble elements in the south of Beijing may be due to the nonlinear relationship between DTT consumption and element concentration (Charrier and Anastasio, 2012; Wu et al., 2022a). As shown in Figure S5, the relationship between most soluble trace elements and $DTT_v$ was more non-linear than linear. As the concentration of soluble elements increases, the growth rate of $DTT_v$ obviously decreases.

In addition to being associated with individual water-soluble species, the interactions between metals and organic compounds also affect the consumption of DTT (Xiong et al., 2017; Wu et al., 2022b), with both synergistic and antagonistic effects. For example, Wu et al. (2022b) measured the DTT consumption of Fe(III) and Cu(II) interacting with 1,4-naphthoquinone, 9,10-phenanthraquinone, citric acid, and 4-nitrocatechol, respectively. Their results showed that Cu(II) had antagonistic effects in interacting with most organics except for citric acid, and Fe(III) had an additive effect on DTT consumption of 1,4-naphthoquinone and citric acid, while it had an antagonistic effect on 1,4-naphthoquinone and 9,10-phenanthraquinone. Due to the complex composition of water-soluble organic aerosols, the knowledge about the effects of organics and metal-organic interactions on DTT activity are still limited, especially the effects of BrC chromophores and their interactions with metals.

**3.3 Sources of DTT activity**

This study analyzed eight organic markers (including levoglucosan, mannosan, and galactosan for biomass burning, hopanes for vehicle emissions, picene for coal combustion, and phthalic acid, isophthalic acid and terephthalic acid for secondary formation) to help identify the sources of DTT activity. The average concentrations of these organic markers are shown in Table S2. The correlation coefficients between

DTT$_v$ and organic markers are shown in Figure S6. In the south, levoglucosan,
mannosan, galactosan, and hopanes had moderate correlation with DTT$_v$ (r of 0.41 to
0.48); phthalic acid, isophthalic acid and terephthalic acid had low to moderate
correlation with DTT$_v$ (r of 0.28 to 0.54); picene had low correlation with DTT$_v$ (r of
0.21). These results suggest that biomass burning and vehicle emissions could have
significant contribution to water-soluble PM$_{2.5}$ OP in the south. In the north, hopanes
had the highest correlation with DTT$_v$ (r = 0.70), indicating that vehicle emissions
could have an important contribution. Levoglucosan, mannosan, galactosan, phthalic
acid, isophthalic acid, terephthalic acid, and picene had moderate to high correlations
with DTT$_v$ in the north, suggesting that biomass and coal burning, and secondary
formation may also have certain contribution to water-soluble PM$_{2.5}$ OP.
To further quantify the sources of DTT activity in the south and the north of
Beijing, the PMF model, which was widely used for the source apportionment of
PM$_{2.5}$ OP (Liu et al., 2018; Shen et al., 2022; Cui et al., 2023), was applied. The input
species include DTT$_v$, soluble elements and organic markers, and five to seven factors
were examined. Due to the oil factor mixed with vehicle emissions factor in the five-
factor solution, and there was no new reasonable factor when increasing the factor
number to seven in the PMF analysis (Figure S7). Finally, six factors were resolved
and quantified using PMF model in the south and north of Beijing, including biomass
burning, coal burning, traffic-related, dust, oil combustion, and secondary formation,
and the profiles of these sources are shown in Figure S8. The uncertainties of PMF
analysis for these sources were 2-14%. Factor 1 is characterized by high contribution
of levoglucosan, mannosan, and galactosan, mainly from biomass burning (Huang et
al., 2014; Chow et al., 2022). The DTT activity of biomass burning organic aerosol
was measured by Wong et al. (2019), which was 48 ± 6 pmol min$^{-1}$ μg$^{-1}$ of WSOC.
Liu et al. (2018) quantified the sources of DTT$_v$ in coastal cities (Jinzhou, Tianjin, and
Yantai) in China with PMF model and multiple linear regression method, and the
results showed that biomass burning contributed 28% on average in winter. Factor 2
exhibits a large fraction of picene, Zn, Mn, Cd, As, and Pb, which is considered to be
coal burning (Huang et al., 2014; Huang et al., 2018). Joo et al. (2018) measured the
DTT activity of $PM_{2.5}$ emitted from coal combustion at different temperatures, with
the highest values of $26 \pm 21$ pmol $min^{-1}$ $\mu g^{-1}$ and $0.10 \pm 0.06$ nmol $min^{-1}$ $m^{-3}$
occurring at 550 °C. Factor 3 is identified as traffic-related emissions, which is
characterized by the higher loading of hopanes, Ba, Sr, Cu and Ni (Huang et al., 2018;
Chow et al., 2022). Vreeland et al. (2017) measured the DTT activity of $PM_{2.5}$ emitted
by side street and highway vehicles in Atlanta, with values of $0.78 \pm 0.60$ nmol $min^{-1}$
$m^{-3}$ and $1.1 \pm 0.60$ nmol $min^{-1}$ $m^{-3}$, respectively. Ting et al. (2023) reported that the
DTT activity of $PM_{2.5}$ from vehicle emissions in Ziqing tunnel in Taiwan, China, was
$0.15$-$0.46$ nmol $min^{-1}$ $m^{-3}$. Factor 4, secondary formation, which is identified by high
levels of phthalic acid, isophthalic acid, and terephthalic acid (Al-Naiema and Stone,
2017; Wang et al., 2020a). Verma et al. (2014) reported that secondary formation
contributed about 30% to the water-soluble DTT activity of $PM_{2.5}$ in urban Atlanta. It
is worth noting that the DTT activity of SOA generated from different precursors is
different (Tuet et al., 2017; Tong et al., 2018). For example, the DTT activity of SOA
from naphthalene was higher than that from isoprene (Tuet et al., 2017; Tong et al.,
2018). Factor 5 is dominated by crustal elements Fe and Ti, mainly from dust (Huang
et al., 2018). The DTT activity of atmospheric particulate matter during dust periods
were reported in previous studies (Chirizzi et al., 2017; Khoshnamvand et al., 2023)
and it has a low contribution in this study. Factor 6 is identified as oil combustion
because of the high levels of V and Ni (Moreno et al., 2011; Minguillón et al., 2014;
Huang et al., 2018).
The source contributions of $DTT_v$ in the south and north of Beijing are shown in
Figure 5, exhibiting obvious district differences. In the south, traffic-related emissions
(39%) and biomass burning (25%) had the most contribution to $DTT_v$, followed by
secondary formation (17%), coal burning (15%), dust (2%), and oil combustion (2%).
In the north, traffic-related emissions (52%) had the highest contribution to $DTT_v$,
followed by coal burning (20%), secondary formation (13%), biomass burning (8%),
oil combustion (4%), and dust (3%). The absolute contribution of each source to
$DTT_v$ varies by 1.2-3.4 times between the south and north of Beijing (Table S3). The
large district differences in sources of $DTT_v$ of water-soluble $PM_{2.5}$ call for more
research on the relationship between sources, chemical composition, formation
processes and OP of $PM_{2.5}$.

**4 Conclusions**
In this study, the water-soluble OP of ambient $PM_{2.5}$ collected in winter in the
south and north of Beijing were quantified, together with the concentration and light
absorption of WSOC, and concentrations of 7 light-absorbing NACs and 14 trace
elements. The average $DTT_v$ value was comparable in the south (3.9 ± 0.9 nmol min$^{-1}$
m$^{-3}$) and north (3.5 ± 1.2 nmol min$^{-1}$ m$^{-3}$), while the $DTT_m$ was higher in the north (65
± 28 pmol min$^{-1}$ μg$^{-1}$) than in the south (36 ± 14 pmol min$^{-1}$ μg$^{-1}$), indicating that the
exposure-relevant OP of water-soluble components of $PM_{2.5}$ was similar in the two
sites and that the intrinsic OP of water-soluble components of $PM_{2.5}$ was higher in the
north than in the south. The correlation between $DTT_v$ and soluble elements was
higher than that between $DTT_v$ and total elements in both the south and north. In the
north, the $DTT_v$ was strongly correlated with soluble Mn, Co, Ni, Zn, As, Cd and Pb
(r > 0.7), and in the south it positively correlated with Mn, Fe, Cr, Co, As and Pb (r >
0.5). In addition, in the north the $DTT_v$ was also positively correlated with WSOC,
$Abs_{365}$ and NACs (r of 0.56 to 0.79), while in the south it was weakly correlated (r ≤
0.4). These results indicate that in the north trace elements and water-soluble organic
compounds, especially BrC chromophores, both had significant contributions to DTT
consumption, and in the south the consumption of DTT may be mainly from trace
elements. Six sources of $DTT_v$ were resolved with the PMF model, including biomass
burning, coal burning, traffic-related, dust, oil combustion, and secondary formation.
On average, traffic-related emissions (39%) and biomass burning (25%) were the
major contributors of $DTT_v$ in the south, and traffic-related emissions (52%) was the
predominated source in the north. The differences in $DTT_v$ sources in the south and
north of Beijing suggest that the relationship between source emissions and

atmospheric processes and PM$_{2.5}$ OP deserve further exploration in order to better understand the regional differences of health impacts of PM$_{2.5}$. Besides, in order to gain a more comprehensive understanding of the regional differences in PM$_{2.5}$ OP, sources and its relationship with chemical composition, longer periods and different seasonal datasets are also need to be studied in the future.

**Date availability.** Raw data used in this study can be obtained from the following open link: https://doi.org/10.5281/zenodo.10791126 (Yuan et al., 2024). It is also available on request by contacting the corresponding authors.

**Supplement.** The supplement related to this article is available online.

**Author contributions.** RJH designed the study. Data analysis was done by WY, CL, LY, HY and RJH. WY, CL, LY, HY and RJH interpreted data, prepared the display items and wrote the manuscript. All authors commented on and discussed the manuscript.

**Competing interests.** The authors declare that they have no conflict of interest.

**Acknowledgements.** We are very grateful to the National Natural Science Foundation of China (NSFC) under Grant No. 41925015, the Strategic Priority Research Program of Chinese Academy of Sciences (XDB40000000), the Key Research Program of Frontier Sciences from the Chinese Academy of Sciences (ZDBS-LY-DQC001), the New Cornerstone Science Foundation through the XPLORER PRIZE, and the Postdoctoral Fellowship Program of CPSF (no. GZC20232628) supported this study.

**Financial support.** This work was supported by the National Natural Science

Foundation of China (NSFC) under Grant No. 41925015, the Strategic Priority Research Program of Chinese Academy of Sciences (XDB40000000), the Key Research Program of Frontier Sciences from the Chinese Academy of Sciences (ZDBS-LY-DQC001), the New Cornerstone Science Foundation through the XPLORER PRIZE, and the Postdoctoral Fellowship Program of CPSF (no. GZC20232628).

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

Contributions from Chemical Components, Toxics, 11,
10.3390/toxics11010059, 2023.
Burnett, R., Chen, H., Szyszkowicz, M., Fann, N., Hubbell, B., Pope, C. A., 3rd, Apte,
J. S., Brauer, M., Cohen, A., Weichenthal, S., Coggins, J., Di, Q., Brunekreef,
B., Frostad, J., Lim, S. S., Kan, H., Walker, K. D., Thurston, G. D., Hayes, R.
B., Lim, C. C., Turner, M. C., Jerrett, M., Krewski, D., Gapstur, S. M., Diver,
W. R., Ostro, B., Goldberg, D., Crouse, D. L., Martin, R. V., Peters, P., Pinault,
L., Tjepkema, M., van Donkelaar, A., Villeneuve, P. J., Miller, A. B., Yin, P.,
Zhou, M., Wang, L., Janssen, N. A. H., Marra, M., Atkinson, R. W., Tsang, H.,
Quoc Thach, T., Cannon, J. B., Allen, R. T., Hart, J. E., Laden, F., Cesaroni, G.,
Forastiere, F., Weinmayr, G., Jaensch, A., Nagel, G., Concin, H., and Spadaro,
J. V.: Global estimates of mortality associated with long-term exposure to
outdoor fine particulate matter, Proc. Natl. Acad. Sci. U. S. A., 115, 9592-9597,
10.1073/pnas.1803222115, 2018.
Calas, A., Uzu, G., Kelly, F. J., Houdier, S., Martins, J. M. F., Thomas, F., Molton, F.,
Charron, A., Dunster, C., Oliete, A., Jacob, V., Besombes, J.-L., Chevrier, F.,
and Jaffrezo, J.-L.: Comparison between five acellular oxidative potential
measurement assays performed with detailed chemistry on $PM_{10}$ samples from
the city of Chamonix (France), Atmos. Chem. Phys., 18, 7863-7875,
10.5194/acp-18-7863-2018, 2018.
Campbell, S. J., Wolfer, K., Utinger, B., Westwood, J., Zhang, Z. H., Bukowiecki, N.,
Steimer, S. S., Vu, T. V., Xu, J., Straw, N., Thomson, S., Elzein, A., Sun, Y.,
Liu, D., Li, L., Fu, P., Lewis, A. C., Harrison, R. M., Bloss, W. J., Loh, M.,
Miller, M. R., Shi, Z., and Kalberer, M.: Atmospheric conditions and
composition that influence $PM_{2.5}$ oxidative potential in Beijing, China, Atmos.
Chem. Phys., 21, 5549-5573, 10.5194/acp-21-5549-2021, 2021.
Cao, T., Li, M., Zou, C., Fan, X., Song, J., Jia, W., Yu, C., Yu, Z., and Peng, P. a.:
Chemical composition, optical properties, and oxidative potential of water-
and methanol-soluble organic compounds emitted from the combustion of

biomass materials and coal, Atmos. Chem. Phys., 21, 13187-13205, 10.5194/acp-21-13187-2021, 2021.

Charrier, J. G. and Anastasio, C.: On dithiothreitol (DTT) as a measure of oxidative potential for ambient particles: evidence for the importance of soluble transition metals, Atmos. Chem. Phys., 12, 9321-9333, 10.5194/acp-12-9321-2012, 2012.

Charrier, J. G., McFall, A. S., Vu, K. K.-T., Baroi, J., Olea, C., Hasson, A., and Anastasio, C.: A Bias in the "Mass-Normalized" DTT Response-An Effect of Non-Linear Concentration Response Curves for Copper and Manganese, Atmos. Environ., 144, 325-334, 2016.

Chen, K., Xu, J., Famiyeh, L., Sun, Y., Ji, D., Xu, H., Wang, C., Metcalfe, S. E., Betha, R., Behera, S. N., Jia, C., Xiao, H., and He, J.: Chemical constituents, driving factors, and source apportionment of oxidative potential of ambient fine particulate matter in a Port City in East China, J. Hazard. Mater., 440, 10.1016/j.jhazmat.2022.129864, 2022.

Chen, Q., Wang, M., Wang, Y., Zhang, L., Li, Y., and Han, Y.: Oxidative Potential of Water-Soluble Matter Associated with Chromophoric Substances in $PM_{2.5}$ over Xi'an, China, Environ. Sci. Technol., 53, 8574-8584, 10.1021/acs.est.9b01976, 2019.

Chirizzi, D., Cesari, D., Guascito, M. R., Dinoi, A., Giotta, L., Donateo, A., and Contini, D.: Influence of Saharan dust outbreaks and carbon content on oxidative potential of water-soluble fractions of $PM_{2.5}$ and $PM_{10}$, Atmos. Environ., 163, 1-8, 10.1016/j.atmosenv.2017.05.021, 2017.

Chow, W. S., Huang, X. H. H., Leung, K. F., Huang, L., Wu, X., and Yu, J. Z.: Molecular and elemental marker-based source apportionment of fine particulate matter at six sites in Hong Kong, China, Sci. Total Environ., 813, 152652, 10.1016/j.scitotenv.2021.152652, 2022.

Chowdhury, P. H., He, Q., Carmieli, R., Li, C., Rudich, Y., and Pardo, M.: Connecting the Oxidative Potential of Secondary Organic Aerosols with Reactive Oxygen

Species in Exposed Lung Cells, Environ. Sci. Technol., 53, 13949-13958, 10.1021/acs.est.9b04449, 2019.

Cui, Y., Zhu, L., Wang, H., Zhao, Z., Ma, S., and Ye, Z.: Characteristics and Oxidative Potential of Ambient $PM_{2.5}$ in the Yangtze River Delta Region: Pollution Level and Source Apportionment, Atmosphere, 14, 10.3390/atmos14030425, 2023.

Daellenbach, K. R., Uzu, G., Jiang, J., Cassagnes, L. E., Leni, Z., Vlachou, A., Stefenelli, G., Canonaco, F., Weber, S., Segers, A., Kuenen, J. J. P., Schaap, M., Favez, O., Albinet, A., Aksoyoglu, S., Dommen, J., Baltensperger, U., Geiser, M., El Haddad, I., Jaffrezo, J. L., and Prevot, A. S. H.: Sources of particulate-matter air pollution and its oxidative potential in Europe, Nature, 587, 414-419, 10.1038/s41586-020-2902-8, 2020.

Fan, X., Li, M., Cao, T., Cheng, C., Li, F., Xie, Y., Wei, S., Song, J., and Peng, P. a.: Optical properties and oxidative potential of water- and alkaline-soluble brown carbon in smoke particles emitted from laboratory simulated biomass burning, Atmos. Environ., 194, 48-57, 10.1016/j.atmosenv.2018.09.025, 2018.

Fang, T., Verma, V., Bates, J. T., Abrams, J., Klein, M., Strickland, M. J., Sarnat, S. E., Chang, H. H., Mulholland, J. A., Tolbert, P. E., Russell, A. G., and Weber, R. J.: Oxidative potential of ambient water-soluble $PM_{2.5}$ in the southeastern United States: contrasts in sources and health associations between ascorbic acid (AA) and dithiothreitol (DTT) assays, Atmos. Chem. Phys., 16, 3865-3879, 10.5194/acp-16-3865-2016, 2016.

Feng, R., Xu, H., Gu, Y., Wang, Z., Han, B., Sun, J., Liu, S., Lu, H., Ho, S. S. H., Shen, Z., and Cao, J.: Variations of Personal Exposure to Particulate Nitrated Phenols from Heating Energy Renovation in China: The First Assessment on Associated Toxicological Impacts with Particle Size Distributions, Environ. Sci. Technol., 56, 3974−3983, 2022.

Gao, D., Fang, T., Verma, V., Zeng, L., and Weber, R. J.: A method for measuring total aerosol oxidative potential (OP) with the dithiothreitol (DTT) assay and comparisons between an urban and roadside site of water-soluble and total OP,

Atmos. Meas. Tech., 10, 2821-2835, 10.5194/amt-10-2821-2017, 2017.
Guascito, M. R., Lionetto, M. G., Mazzotta, F., Conte, M., Giordano, M. E., Caricato,
R., De Bartolomeo, A. R., Dinoi, A., Cesari, D., Merico, E., Mazzotta, L., and
Contini, D.: Characterisation of the correlations between oxidative potential
and in vitro biological effects of $PM_{10}$ at three sites in the central
Mediterranean,      J.      Hazard.      Mater.,      448,      130872,
10.1016/j.jhazmat.2023.130872, 2023.
Hecobian, A., Zhang, X., Zheng, M., Frank, N., Edgerton, E. S., and Weber, R. J.:
Water-Soluble    Organic    Aerosol    material    and    the    light-absorption
characteristics of aqueous extracts measured over the Southeastern United
States, Atmos. Chem. Phys., 10, 5965-5977, 10.5194/acp-10-5965-2010, 2010.
Ho, K. F., Ho, S. S. H., Huang, R.-J., Liu, S. X., Cao, J.-J., Zhang, T., Chuang, H.-C.,
Chan, C. S., Hu, D., and Tian, L.: Characteristics of water-soluble organic
nitrogen in fine particulate matter in the continental area of China, Atmos.
Environ., 106, 252-261, 10.1016/j.atmosenv.2015.02.010, 2015.
Huang, R. J., Cheng, R., Jing, M., Yang, L., Li, Y., Chen, Q., Chen, Y., Yan, J., Lin, C.,
Wu, Y., Zhang, R., El Haddad, I., Prevot, A. S. H., O'Dowd, C. D., and Cao, J.:
Source-Specific Health Risk Analysis on Particulate Trace Elements: Coal
Combustion and Traffic Emission As Major Contributors in Wintertime
Beijing, Environ. Sci. Technol., 52, 10967-10974, 10.1021/acs.est.8b02091,

2018.

Huang, R. J., Yang, L., Shen, J., Yuan, W., Gong, Y., Guo, J., Cao, W., Duan, J., Ni, H.,
Zhu, C., Dai, W., Li, Y., Chen, Y., Chen, Q., Wu, Y., Zhang, R., Dusek, U.,
O'Dowd, C., and Hoffmann, T.: Water-Insoluble Organics Dominate Brown
Carbon in Wintertime Urban Aerosol of China: Chemical Characteristics and
Optical    Properties,    Environ.    Sci.    Technol.,    54,    7836-7847,
10.1021/acs.est.0c01149, 2020.
Huang, R. J., Zhang, Y., Bozzetti, C., Ho, K. F., Cao, J. J., Han, Y., Daellenbach, K. R.,
Slowik, J. G., Platt, S. M., Canonaco, F., Zotter, P., Wolf, R., Pieber, S. M.,

Bruns, E. A., Crippa, M., Ciarelli, G., Piazzalunga, A., Schwikowski, M., Abbaszade, G., Schnelle-Kreis, J., Zimmermann, R., An, Z., Szidat, S., Baltensperger, U., El Haddad, I., and Prevot, A. S.: High secondary aerosol contribution to particulate pollution during haze events in China, Nature, 514, 218-222, 10.1038/nature13774, 2014.

Jiang, H., Xie, Y., Ge, Y., He, H., and Liu, Y.: Effects of ultrasonic treatment on dithiothreitol (DTT) assay measurements for carbon materials, J. Environ. Sci., 84, 51–58, 2019.

Joo, H. S., Batmunkh, T., Borlaza, L. J. S., Park, M., Lee, K. Y., Lee, J. Y., Chang, Y. W., and Park, K.: Physicochemical properties and oxidative potential of fine particles produced from coal combustion, Aerosol Sci. Technol., 52, 1134-1144, 10.1080/02786826.2018.1501152, 2018.

Khoshnamvand, N., Nodehi, R. N., Hassanvand, M. S., and Naddafi, K.: Comparison between oxidative potentials measured of water-soluble components in ambient air $PM_1$ and $PM_{2.5}$ of Tehran, Iran, Air Qual. Atmos. Hlth., 16, 1311-1320, 10.1007/s11869-023-01343-y, 2023.

Laskin, A., Laskin, J., and Nizkorodov, S. A.: Chemistry of atmospheric brown carbon, Chem. Rev., 115, 4335-4382, 10.1021/cr5006167, 2015.

Lelieveld, S., Wilson, J., Dovrou, E., Mishra, A., Lakey, P. S. J., Shiraiwa, M., Poschl, U., and Berkemeier, T.: Hydroxyl Radical Production by Air Pollutants in Epithelial Lining Fluid Governed by Interconversion and Scavenging of Reactive Oxygen Species, Environ. Sci. Technol., 55, 14069-14079, 10.1021/acs.est.1c03875, 2021.

Lin, P., Bluvshtein, N., Rudich, Y., Nizkorodov, S. A., Laskin, J., and Laskin, A.: Molecular chemistry of atmospheric brown carbon inferred from a nationwide biomass burning event, Environ. Sci. Technol., 51, 11561–11570, 2017.

Lionetto, M., Guascito, M., Giordano, M., Caricato, R., De Bartolomeo, A., Romano, M., Conte, M., Dinoi, A., and Contini, D.: Oxidative Potential, Cytotoxicity, and Intracellular Oxidative Stress Generating Capacity of $PM_{10}$: A Case Study

in South of Italy, Atmosphere, 12, 10.3390/atmos12040464, 2021.

Liu, W., Xu, Y., Liu, W., Liu, Q., Yu, S., Liu, Y., Wang, X., and Tao, S.: Oxidative

potential of ambient $PM_{2.5}$ in the coastal cities of the Bohai Sea, northern

China: Seasonal variation and source apportionment, Environ. Pollut., 236,

514-528, 10.1016/j.envpol.2018.01.116, 2018.

Liu, Y., Yan, C. Q., Ding, X., Wang, X. M., Fu, Q. Y., Zhao, Q. B., Zhang, Y. H., Duan,

Y. S., Qiu, X. H., and Zheng, M.: Sources and spatial distribution of

particulate polycyclic aromatic hydrocarbons in Shanghai, China, Sci. Total

Environ., 584-585, 307-317, https://doi.org/10.1016/j.scitotenv.2016.12.134,

2017.

648    Ma, X., Nie, D., Chen, M., Ge, P., Liu, Z., Ge, X., Li, Z., and Gu, R.: The Relative

Contributions of Different Chemical Components to the Oxidative Potential of

Ambient Fine Particles in Nanjing Area, Int. J. Environ. Res. Public Health, 18,

2789, 10.3390/ijerph18062789, 2021.

Miljevic, B., Hedayat, F., Stevanovic, S., Fairfull-Smith, K. E., Bottle, S. E., and

Ristovski, Z. D.: To sonicate or not to sonicate PM filters: reactive oxygen

species generation upon ultrasonic irradiation, Aerosol. Sci. Technol., 48,

1276-1284, 2014.

Minguillón, M. C., Cirach, M., Hoek, G., Brunekreef, B., Tsai, M., de Hoogh, K.,

Jedynska, A., Kooter, I. M., Nieuwenhuijsen, M., and Querol, X.: Spatial

variability of trace elements and sources for improved exposure assessment in

Barcelona, Atmos. Environ., 89, 268-281, 10.1016/j.atmosenv.2014.02.047,

2014.

Moreno, T., Querol, X., Alastuey, A., Reche, C., Cusack, M., Amato, F., Pandolfi, M.,

Pey, J., Richard, A., Prévôt, A. S. H., Furger, M., and Gibbons, W.: Variations

in time and space of trace metal aerosol concentrations in urban areas and their

surroundings, Atmos. Chem. Phys., 11, 9415-9430, 10.5194/acp-11-9415-2011,

2011.

Oh, S. H., Park, K., Park, M., Song, M., Jang, K. S., Schauer, J. J., Bae, G. N., and

Bae, M. S.: Comparison of the sources and oxidative potential of $PM_{2.5}$ during winter time in large cities in China and South Korea, Sci. Total Environ., 859, 160369, 10.1016/j.scitotenv.2022.160369, 2023.

Paatero, P.: Least squares formation of robust non negative factor analysis, Chemometr. Intell. Lab., 37, 23-35, 1997.

Puthussery, J. V., Dave, J., Shukla, A., Gaddamidi, S., Singh, A., Vats, P., Salana, S., Ganguly, D., Rastogi, N., Tripathi, S. N., and Verma, V.: Effect of Biomass Burning, Diwali Fireworks, and Polluted Fog Events on the Oxidative Potential of Fine Ambient Particulate Matter in Delhi, India, Environ. Sci. Technol., 56, 14605-14616, 10.1021/acs.est.2c02730, 2022.

Saffari, A., Daher, N., Shafer, M. M., Schauer, J. J., and Sioutas, C.: Global perspective on the oxidative potential of airborne particulate matter: a synthesis of research findings, Environ. Sci. Technol., 48, 7576-7583, 10.1021/es500937x, 2014.

Shafer, M. M., Hemming, J. D., Antkiewicz, D. S., and Schauer, J. J.: Oxidative potential of size-fractionated atmospheric aerosol in urban and rural sites across Europe, Faraday Discuss., 189, 381-405, 10.1039/c5fd00196j, 2016.

Shen, J., Taghvaee, S., La, C., Oroumiyeh, F., Liu, J., Jerrett, M., Weichenthal, S., Del Rosario, I., Shafer, M. M., Ritz, B., Zhu, Y., and Paulson, S. E.: Aerosol Oxidative Potential in the Greater Los Angeles Area: Source Apportionment and Associations with Socioeconomic Position, Environ. Sci. Technol., 56, 17795-17804, 10.1021/acs.est.2c02788, 2022.

Ting, Y. C., Chang, P. K., Hung, P. C., Chou, C. C., Chi, K. H., and Hsiao, T. C.: Characterizing emission factors and oxidative potential of motorcycle emissions in a real-world tunnel environment, Environ. Res., 234, 116601, 10.1016/j.envres.2023.116601, 2023.

Tong, H., Lakey, P. S. J., Arangio, A. M., Socorro, J., Kampf, C. J., Berkemeier, T., Brune, W. H., Poschl, U., and Shiraiwa, M.: Reactive oxygen species formed in aqueous mixtures of secondary organic aerosols and mineral dust

influencing cloud chemistry and public health in the Anthropocene, Faraday Discuss., 200, 251-270, 10.1039/c7fd00023e, 2017.

Tong, H., Lakey, P. S. J., Arangio, A. M., Socorro, J., Shen, F., Lucas, K., Brune, W. H., Poschl, U., and Shiraiwa, M.: Reactive Oxygen Species Formed by Secondary Organic Aerosols in Water and Surrogate Lung Fluid, Environ. Sci. Technol., 52, 11642-11651, 10.1021/acs.est.8b03695, 2018.

Tuet, W. Y., Chen, Y., Xu, L., Fok, S., Gao, D., Weber, R. J., and Ng, N. L.: Chemical oxidative potential of secondary organic aerosol (SOA) generated from the photooxidation of biogenic and anthropogenic volatile organic compounds, Atmos. Chem. Phys., 17, 839-853, 10.5194/acp-17-839-2017, 2017.

Tuet, W. Y., Liu, F., de Oliveira Alves, N., Fok, S., Artaxo, P., Vasconcellos, P., Champion, J. A., and Ng, N. L.: Chemical Oxidative Potential and Cellular Oxidative Stress from Open Biomass Burning Aerosol, Environ. Sci. Technol. Lett., 6, 126-132, 10.1021/acs.estlett.9b00060, 2019.

Verma, V., Fang, T., Xu, L., Peltier, R. E., Russell, A. G., Ng, N. L., and Weber, R. J.: Organic aerosols associated with the generation of reactive oxygen species (ROS) by water-soluble $PM_{2.5}$, Environ. Sci. Technol., 49, 4646-4656, 10.1021/es505577w, 2015.

Verma, V., Rico-Martinez, R., Kotra, N., King, L., Liu, J., Snell, T. W., and Weber, R. J.: Contribution of water-soluble and insoluble components and their hydrophobic/hydrophilic subfractions to the reactive oxygen species-generating potential of fine ambient aerosols, Environ. Sci. Technol., 46, 11384-11392, 10.1021/es302484r, 2012.

Verma, V., Fang, T., Guo, H., King, L., Bates, J. T., Peltier, R. E., Edgerton, E., Russell, A. G., and Weber, R. J.: Reactive oxygen species associated with water-soluble $PM_{2.5}$ in the southeastern United States: spatiotemporal trends and source apportionment, Atmos. Chem. Phys., 14, 12915-12930, 10.5194/acp-14-12915-2014, 2014.

Vreeland, H., Weber, R., Bergin, M., Greenwald, R., Golan, R., Russell, A. G., Verma,

V., and Sarnat, J. A.: Oxidative potential of $PM_{2.5}$ during Atlanta rush hour: Measurements of in-vehicle dithiothreitol (DTT) activity, Atmos. Environ., 165, 169-178, 10.1016/j.atmosenv.2017.06.044, 2017.

Wang, J., Lin, X., Lu, L., Wu, Y., Zhang, H., Lv, Q., Liu, W., Zhang, Y., and Zhuang, S.: Temporal variation of oxidative potential of water soluble components of ambient $PM_{2.5}$ measured by dithiothreitol (DTT) assay, Sci. Total Environ., 649, 969-978, 10.1016/j.scitotenv.2018.08.375, 2019.

Wang, T., Huang, R. J., Li, Y., Chen, Q., Chen, Y., Yang, L., Guo, J., Ni, H., Hoffmann, T., Wang, X., and Mai, B.: One-year characterization of organic aerosol markers in urban Beijing: Seasonal variation and spatiotemporal comparison, Sci. Total Environ., 743, 140689, 10.1016/j.scitotenv.2020.140689, 2020a.

Wang, Y., Wang, M., Li, S., Sun, H., Mu, Z., Zhang, L., Li, Y., and Chen, Q.: Study on the oxidation potential of the water-soluble components of ambient $PM_{2.5}$ over Xi'an, China: Pollution levels, source apportionment and transport pathways, Environ. Int., 136, 105515, 10.1016/j.envint.2020.105515, 2020b.

Wong, J. P. S., Tsagkaraki, M., Tsiodra, I., Mihalopoulos, N., Violaki, K., Kanakidou, M., Sciare, J., Nenes, A., and Weber, R. J.: Effects of Atmospheric Processing on the Oxidative Potential of Biomass Burning Organic Aerosols, Environ. Sci. Technol., 53, 6747-6756, 10.1021/acs.est.9b01034, 2019.

Wu, N., Lu, B., Chen, Q., Chen, J., and Li, X.: Connecting the Oxidative Potential of Fractionated Particulate Matter With Chromophoric Substances, J. Geophys. Res-Atmos., 127, 10.1029/2021jd035503, 2022a.

Wu, N., Lyu, Y., Lu, B., Cai, D., Meng, X., and Li, X.: Oxidative potential induced by metal-organic interaction from $PM_{2.5}$ in simulated biological fluids, Sci. Total Environ., 848, 157768, 10.1016/j.scitotenv.2022.157768, 2022b.

Xing, C., Wang, Y., Yang, X., Zeng, Y., Zhai, J., Cai, B., Zhang, A., Fu, T. M., Zhu, L., Li, Y., Wang, X., and Zhang, Y.: Seasonal variation of driving factors of ambient $PM_{2.5}$ oxidative potential in Shenzhen, China, Sci. Total Environ., 862, 160771, 10.1016/j.scitotenv.2022.160771, 2023.

Xiong, Q., Yu, H., Wang, R., Wei, J., and Verma, V.: Rethinking Dithiothreitol-Based Particulate Matter Oxidative Potential: Measuring Dithiothreitol Consumption versus Reactive Oxygen Species Generation, Environ. Sci. Technol., 51, 6507-6514, 10.1021/acs.est.7b01272, 2017.

Yalamanchili, J., Hennigan, C. J., and Reed, B. E.: Measurement artifacts in the dithiothreitol (DTT) oxidative potential assay caused by interactions between aqueous metals and phosphate buffer, J. Hazard. Mater., 456, 131693, 2023.

Yu, H., Wei, J., Cheng, Y., Subedi, K., and Verma, V.: Synergistic and Antagonistic Interactions among the Particulate Matter Components in Generating Reactive Oxygen Species Based on the Dithiothreitol Assay, Environ. Sci. Technol., 52, 2261–2270, 2018.

Yu, Q., Chen, J., Qin, W., Ahmad, M., Zhang, Y., Sun, Y., Xin, K., and Ai, J.: Oxidative potential associated with water-soluble components of $PM_{2.5}$ in Beijing: The important role of anthropogenic organic aerosols, J. Hazard. Mater., 433, 128839, 10.1016/j.jhazmat.2022.128839, 2022a.

Yu, S., Liu, W., Xu, Y., Yi, K., Zhou, M., Tao, S., and Liu, W.: Characteristics and oxidative potential of atmospheric $PM_{2.5}$ in Beijing: Source apportionment and seasonal variation, Sci. Total Environ., 650, 277-287, 10.1016/j.scitotenv.2018.09.021, 2019.

Yu, Y., Sun, Q., Li, T., Ren, X., Lin, L., Sun, M., Duan, J., and Sun, Z.: Adverse outcome pathway of fine particulate matter leading to increased cardiovascular morbidity and mortality: An integrated perspective from toxicology and epidemiology, J. Hazard. Mater., 430, 128368, 10.1016/j.jhazmat.2022.128368, 2022b.

Yu, Y., Cheng, P., Li, Y., Gu, J., Gong, Y., Han, B., Yang, W., Sun, J., Wu, C., Song, W., and Li, M.: The association of chemical composition particularly the heavy metals with the oxidative potential of ambient $PM_{2.5}$ in a megacity (Guangzhou) of southern China, Environ. Res., 213, 113489, 10.1016/j.envres.2022.113489, 2022c.

Yuan, W., Huang, R.-J., Luo, C., Yang, L., Cao, W., Guo, J., and Yang, H.:
Measurement report: Oxidation potential of water-soluble aerosol components
in the southern and northern of Beijing, Zenodo [data set],
https://doi.org/10.5281/zenodo.10791126, 2024.
Yuan, W., Huang, R.-J., Yang, L., Guo, J., Chen, Z., Duan, J., Wang, T., Ni, H., Han,
Y., Li, Y., Chen, Q., Chen, Y., Hoffmann, T., and O'Dowd, C.: Characterization
of the light-absorbing properties, chromophore composition and sources of
brown carbon aerosol in Xi'an, northwestern China, Atmos. Chem. Phys., 20,
5129-5144, 10.5194/acp-20-5129-2020, 2020.
Zhang, Q., Ma, H., Li, J., Jiang, H., Chen, W., Wan, C., Jiang, B., Dong, G., Zeng, X.,
Chen, D., Lu, S., You, J., Yu, Z., Wang, X., and Zhang, G.: Nitroaromatic
Compounds from Secondary Nitrate Formation and Biomass Burning Are
Major Proinflammatory Components in Organic Aerosols in Guangzhou: A
Bioassay Combining High-Resolution Mass Spectrometry Analysis, Environ.
Sci. Technol., 57, 21570-21580, https://doi.org/10.1021/acs.est.3c04983, 2023.
Zheng, Y., Davis, S. J., Persad, G. G., and Caldeira, K.: Climate effects of aerosols
reduce economic inequality, Nat. Clim. Chang., 10, 220-224, 2020.




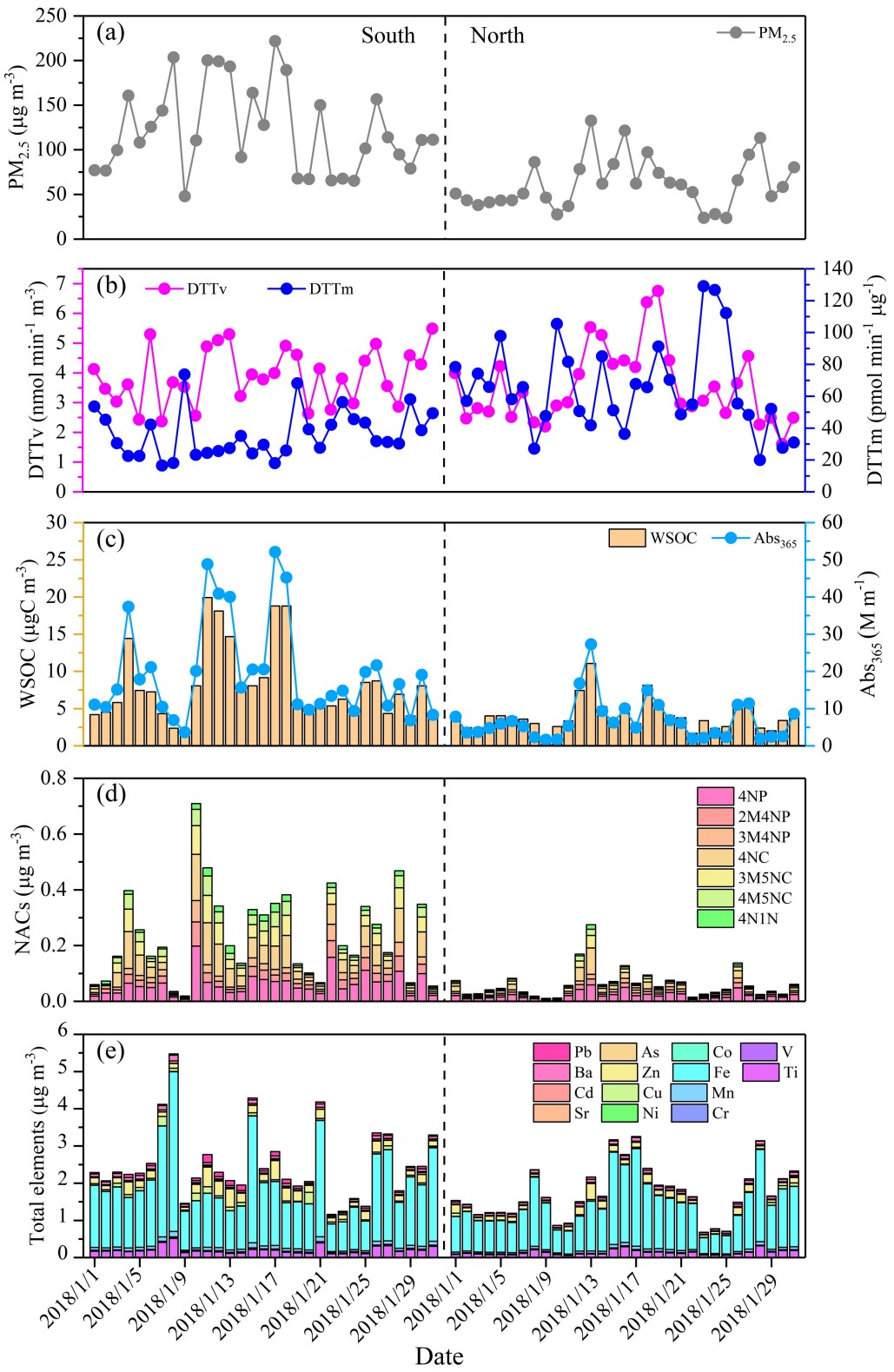


**Figure 1.** Time series of (a) PM$_{2.5}$ concentration, (b) DTT$_v$ and DTT$_m$, (c) concentration and light absorption at wavelength 365 nm (Abs$_{365}$) of WSOC, concentrations of (d) NACs and (e) total elements.

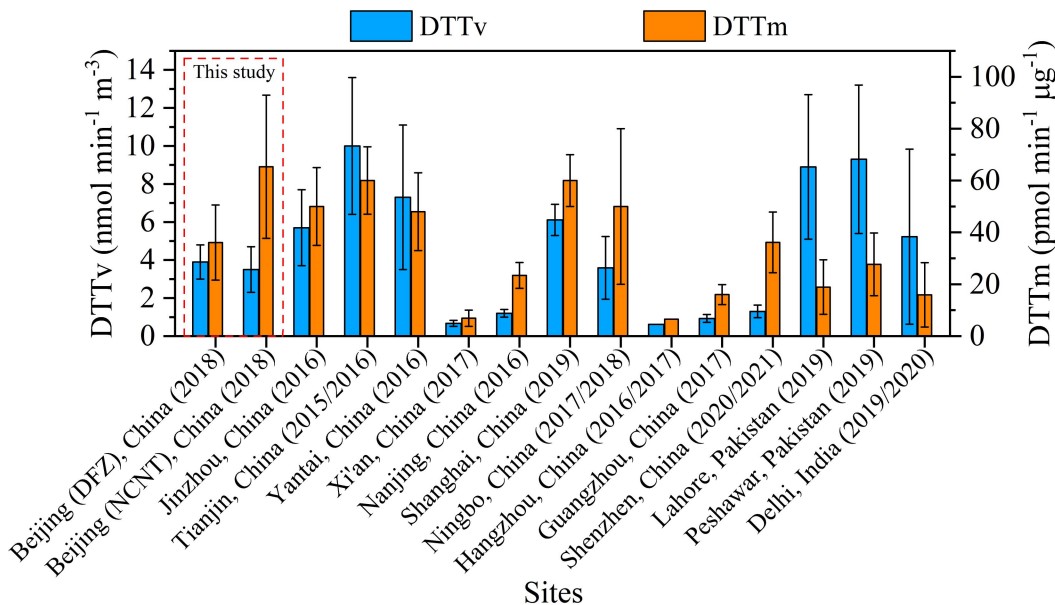


**Figure 2.** Comparison of $DTT_v$ and $DTT_m$ values of water-soluble $PM_{2.5}$ measured in
this study with those measured in other areas of Asia during similar period.

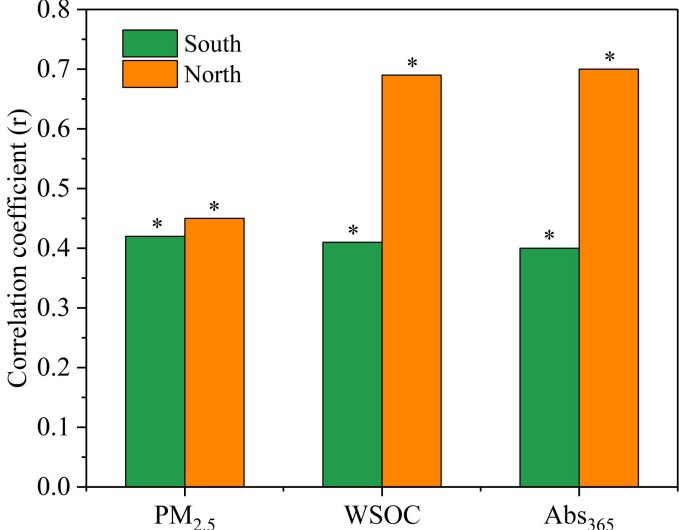


**Figure 3.** Correlation coefficients between $DTT_v$ and $PM_{2.5}$, WSOC, and $Abs_{365}$ in the
south and north of Beijing (* indicates correlation is significant at the 0.05 level).

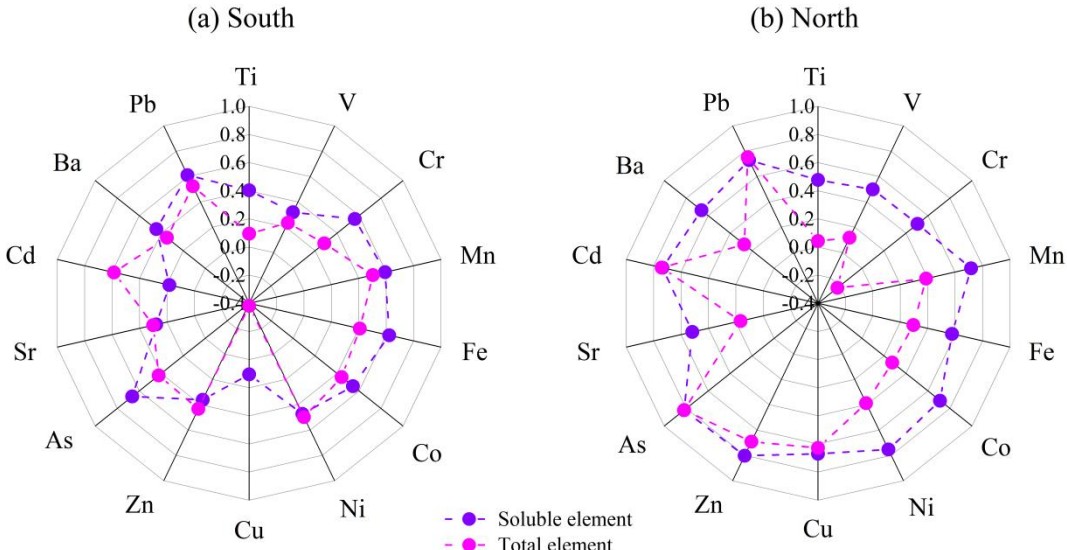


**Figure 4.** Correlation coefficients between $DTT_v$ and elements in the (a) south and (b)
north of Beijing.

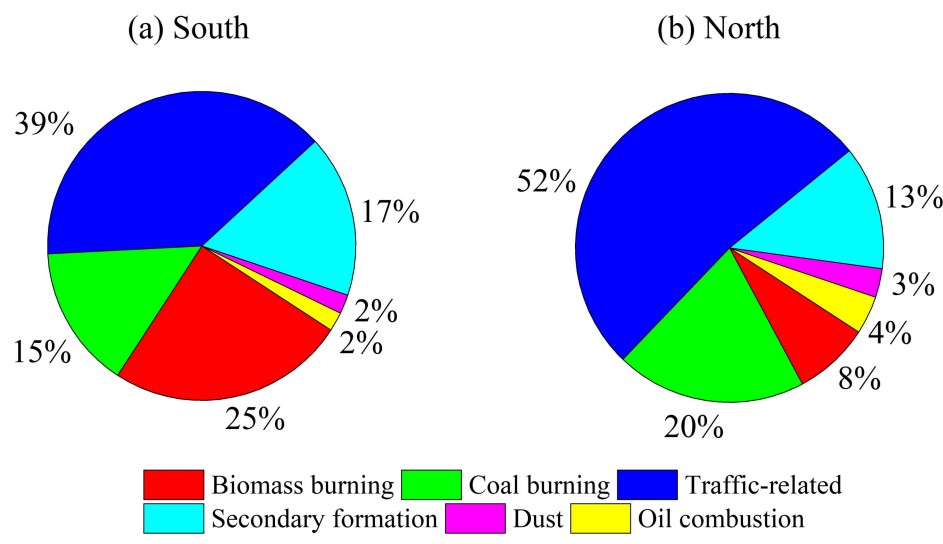


**Figure 5.** Contributions of resolved sources to $DTT_v$ in the (a) south and (b) north of
Beijing.