# Peer review of "Measurement report: Oxidation potential of water-soluble"

_EGUsphere, 2024_

## Author Comment (AC1)

The authors thank the referees to review our manuscript and particularly for the valuable comments and suggestions that have significantly improved the manuscript. We provide below point-by-point responses (in blue) to the referees' comments and have made changes accordingly in the revised manuscript.

Referee #1

Filters were collected at two sites in Beijing, one in what is referred to as south (39.61°N) and the other the north site (39.99°N). Based on comparing measurements of water soluble DTTv and $PM_{2.5}$ mass concentration and various chemical species, this research finds practically all measured parameters were substantially higher in the south vs north. However, DTTv was similar, thus DTTm was much higher in the north; that is the water-soluble components of the particles were concluded to be more toxic at the north site (noted in lines 209-213).

Response: Thanks for pointing this out.

In lines 247-248 of the revised manuscript, we have changed "...indicates that the intrinsic OP of $PM_{2.5}$ was higher in the north than in the south." to "...indicates that the intrinsic OP of water-soluble components of $PM_{2.5}$ was higher in the north than in the south."

In lines 248-250 of the revised manuscript, we have changed "...indicate that the exposure-relevant toxicity of $PM_{2.5}$ was comparable in the two sites" to "...indicate that the exposure-relevant OP of water-soluble components of $PM_{2.5}$ was comparable in the two sites"

In lines 264-266 of the revised manuscript, we have changed "...The differences in $DTT_v$ and $DTT_m$ values in different regions reflect the regional differences in $PM_{2.5}$ exposure risk and intrinsic toxicity, which can be explained by..." to "...The differences in water-soluble DTT activity of $PM_{2.5}$ in different regions can be explained by..."

In lines 407-409 of the revised manuscript, we have changed "...indicating that the $PM_{2.5}$ exposure-relevant toxicity was similar in the two sites and that the $PM_{2.5}$ intrinsic toxicity was higher in the north than in the south." to "...indicating that

the exposure-relevant OP of water-soluble components of $PM_{2.5}$ was similar in the two sites and that the intrinsic OP of water-soluble components of $PM_{2.5}$ was higher in the north than in the south."

The authors perform a correlation and source apportionment analysis and find differences. However, no possible explanation is given for the observed differences in toxicity other than it is likely due to different species. Another possibility is raised is that the nonlinearity in the DTT assay may be having an effect (noted in the manuscript line 271-273). This seems to be a viable explanation since it is known that $DTT_v$ response decreases with increasing concentrations of species in the extraction vial such that at high metals concentrations the assay becomes much less responsive to differences in metals concentrations. Since the metals concentrations are very high in this study, this could explain the similar $DTT_v$ at the two sites and the higher $DTT_m$ at the north site with lower $PM_{2.5}$ mass concentration. Since the difference in $DTT_m$ between the sites is a key finding of this paper and is claimed to indicate a more toxic aerosol in the north site, the possibility that it is instead driven by an artifact should be investigated in detail. There are a number of things that could be done. Redo the analysis at a constant aerosol particles mass in the extraction vial at both sites, as suggested by other investigators (Charrier, J. G., A. S. McFall, K. K. T. Vu, J. Baroi, C. Olea, A. Hasson, and C. Anastasio (2016), A bias in the "mass-normalized" DTT response – An effect of non-linear concentration-response curves for copper and manganese, Atm Env, 325-334). Redo the analysis at different particle masses for both sites and see how much that affects $DTT_m$.

Response: Thanks for the professional comment. We agree that for samples with significant contributions from species whose DTT response is non-linear related to $PM_{2.5}$ mass (e.g., Cu, Mn), the $DTT_m$ response may largely depend on the concentration of $PM_{2.5}$ in the extract, as reported in Charrier et al. (2016). We therefore re-visited the response of $DTT_m$ to $PM_{2.5}$ concentration in the extract using sample with high concentrations of soluble Cu and Mn (Figure R1). Similar to the results observed by Charrier et al. (2016), the $DTT_m$ response

exhibits a non-linear decrease with increasing $PM_{2.5}$ concentrations. It can be seen that in the range of $PM_{2.5}$ concentrations less than 150 µg mL$^{-1}$, the $DTT_m$ response is greatly affected by the $PM_{2.5}$ concentrations. However, when the concentrations of $PM_{2.5}$ are greater than 150 µg mL$^{-1}$, the $DTT_m$ response changes little (< 12%) with the increase of $PM_{2.5}$ concentrations. In this study, the concentrations of $PM_{2.5}$ in the extraction solution of most samples (> 80%) at the two sites are greater than 150 µg mL$^{-1}$ (ranged from 78.7 to 748.7 µg mL$^{-1}$, with an average value of 307.7 ± 167.9 µg mL$^{-1}$), therefore, the difference in $PM_{2.5}$ concentrations in different sample extracts should have a relatively small impact on the difference in $DTT_m$ values of the samples.

In lines 186-199 of the revised manuscript, it now reads "Considering that for samples with significant contributions from species whose DTT response is non-linear related to $PM_{2.5}$ mass (e.g., Cu, Mn), the $DTT_m$ value depends on the concentration of $PM_{2.5}$ in the extraction solution (Charrier et al., 2016). The response of $DTT_m$ to $PM_{2.5}$ concentration in the extraction solution was analyzed using sample with high concentrations of soluble Cu and Mn (Figure S2). In the range of $PM_{2.5}$ concentrations less than 150 µg mL$^{-1}$, the $DTT_m$ response was greatly affected by $PM_{2.5}$ concentrations, however, when the concentrations of $PM_{2.5}$ in the extract were greater than 150 µg mL$^{-1}$, the $DTT_m$ response changed little (< 12%) with the increase of $PM_{2.5}$ concentrations. In this study, the concentrations of $PM_{2.5}$ in the extraction solution of most samples from the two sites were greater than 150 µg mL$^{-1}$ (ranged from 78.7 to 748.7 µg mL$^{-1}$, with average values of 408.9 ± 164.1 and 206.6 ± 95.0 µg mL$^{-1}$ in the south and north, respectively), therefore, the difference in $PM_{2.5}$ concentrations in different sample extracts should had a relatively small impact on the difference in $DTT_m$ values of the samples."

[Figure]

**Figure R1.** Measured $DTT_m$ response as a function of $PM_{2.5}$ concentration in the extraction solution. The Cu and Mn concentrations in this sample are 23.7 and 23.4 ng $m^{-3}$, respectively.

In contrast to this possible limitation with the DTT assay affecting $DTT_m$, the results of Fig 3 showing higher correlations with $Abs_{365}$ in the north and suggesting more influence from NACs, is a possible cause for the higher $DTT_m$ in the north. Maybe this idea could be explored more, e.g., although $Abs_{365}$ is smaller in the north what are the MACs ($Abs_{365}/PM_{2.5}$ mass)? Maybe a similar analysis could be done for NACs? From a rough calculation based on Table S1, the $Abs_{365}$/mass (MAC) at the north site is 0.26 vs 0.16 at the south site. For the sum of NACs/mass, the ratio is about 2.3 at the north site and 2 at the south site, both of these suggesting that the $DTT_m$ could be higher at the north sites due to these organic species, which could maybe be due to a higher proportion of vehicle emissions.

Response: Thanks for your careful reading and suggestions. The mass absorption coefficients (MACs) of water-soluble organic compounds at wavelength of 365 nm ($MAC_{365}$) is $Abs_{365}$/WSOC, the $MAC_{365}$ was $2.4 \pm 0.3$ and $1.5 \pm 0.5$ m$^2$ gC$^{-1}$ in the south and north, respectively. The sum of NACs/WSOC was $3.1 \pm 1.8\%$ and $1.5 \pm 0.6\%$ in the south and north, respectively. These trends were similar to the $Abs_{365}$ and NACs concentrations, which were also higher in the south than in the north, as discussed in Section 3.1 of the manuscript. Therefore, this study

didn't further analyze and discuss the differences in MACs and NACs/WSOC between the two sites.

Overall, this the results of this paper are interesting, but more analysis is needed.

Specific comments.

In Section 2.1 please state: What is the actual distance between sites (I get 42 km). How many samples were collected at each site? State this

Response: Thanks for pointing this out. You are right, the distance between the two sites is about 42 km and a scale bar has been added to Figure S1 (Figure R2). 31 samples were collected at each site.

In lines 110-111 of the revised manuscript, it now reads "...The distance between the two sampling sites is about 42 km."

In line 117 of the revised manuscript, it now reads "...31 samples were collected at each site."

[Figure]

**Figure R2.** Map of the sampling sites. NCNT and DFZ are abbreviations for the north (the National Center for Nanoscience and Technology) and south (the Dingfuzhuang village, Daxing district) sites of Beijing, China, respectively. The left panel from Ministry of Natural Resources of China, and the right panel from Google Maps.

In Fig 2, were the protocols for the DTT analysis for the studies shown the same across all these studies?

Response: Thanks for pointing this out. The studies shown in Figure 2 mainly considered that their sampling time was similar to that of this study, and DTT assay was used. The study in Beijing (CP), China, and Gwangju, Korea didn't described the extraction solvent and method of samples in the DTT assay (Oh et al., 2023). The protocols of the DTT analysis for most of the rest studies were similar. Except for the study in Beijing (IAP), China (Campbell et al., 2021), which extracted the samples with methanol and controlled the concentration of $PM_{2.5}$ in reaction to $\sim 20$ μg, other studies had used a constant filter area for each sample, and extracted with water. Except for the studies in Xi'an (Wang et al., 2020b) and Guangzhou (Yu et al., 2022c), China, which extracted the samples by vortexing and agitation, respectively, and the study in Delhi, India (Puthussery et al., 2022), which measured the DTT activity using their automated online DTT activity measurement instrument, other studies extracted the samples by ultrasonic. A previous study reported that the difference in OP of water-soluble $PM_{2.5}$ measured by DTT assay was little for sample extracted by ultrasonic and shaking (Gao et al., 2017). Therefore, we have now removed the studies in Beijing, China, and Gwangju, Korea (Campbell et al., 2021; Oh et al., 2023) in Figure 2 (Figure R3).

In lines 254-255 of the revised manuscript, we have changed "...the $DTT_v$ values measured in Beijing (Campbell et al., 2021; Oh et al., 2023; this study)" to "...the $DTT_v$ values measured in Beijing in this study..."

In lines 258-259 of the revised manuscript, we have changed "...higher than that in Xi'an, Nanjing, Hangzhou, Guangzhou, and Shenzhen in China, and Gwangju in Korea..." to "...higher than that in Xi'an, Nanjing, Hangzhou, Guangzhou, and Shenzhen in China..."

[Figure]

**Figure R3.** Comparison of $DTT_v$ and $DTT_m$ values of water-soluble $PM_{2.5}$ measured in this study with those measured in other areas of Asia during similar period.

Estimate the mass of $PM_{2.5}$ in the extract that was used for the DTT analysis, show a summary comparing the two sets of data, S and North. (This relates to the comment about possible artifacts related to non-linear response of the assay to metals).

Response: In lines 186-199 of the revised manuscript, it now reads "Considering that for samples with significant contributions from species whose DTT response is non-linear related to $PM_{2.5}$ mass (e.g., Cu, Mn), the $DTT_m$ value depends on the concentration of $PM_{2.5}$ in the extraction solution (Charrier et al., 2016). The response of $DTT_m$ to $PM_{2.5}$ concentration in the extraction solution was analyzed using sample with high concentrations of soluble Cu and Mn (Figure S2). In the range of $PM_{2.5}$ concentrations less than 150 µg mL$^{-1}$, the $DTT_m$ response was greatly affected by $PM_{2.5}$ concentrations, however, when the concentrations of $PM_{2.5}$ in the extract were greater than 150 µg mL$^{-1}$, the $DTT_m$ response changed little (< 12%) with the increase of $PM_{2.5}$ concentrations. In this study, the concentrations of $PM_{2.5}$ in the extraction solution of most samples from the two sites were greater than 150 µg mL$^{-1}$ (ranged from 78.7 to 748.7 µg mL$^{-1}$, with average values of 408.9 ± 164.1 and 206.6 ± 95.0 µg mL$^{-1}$ in the south and north, respectively), therefore, the difference in $PM_{2.5}$ concentrations in different

sample extracts should had a relatively small impact on the difference in $DTT_m$ values of the samples."

In Figure 2, is this comparison just WS DTT for all data shown. Be clear on what is being compared. Does the China data in Figure 2 support the findings of a difference in this paper.

Response: Thanks for the professional comment. We rechecked the comparative studies in Figure 2 and removed the studies in Beijing, China, and Gwangju, Korea (Campbell et al., 2021; Oh et al., 2023), which were not WS DTT. Due to the differences in chemical composition, sources and atmospheric formation processes of water-soluble $PM_{2.5}$ in different regions, the differences in DTT activity in other regions in Figure 2 were not all similar to that in this study. The $DTT_v$ of Xi'an ($0.7 \pm 0.2$ nmol min$^{-1}$ m$^{-3}$) and Guangzhou ($0.9 \pm 0.2$ nmol min$^{-1}$ m$^{-3}$), China was comparable, while the $DTT_m$ of Guangzhou ($16.0 \pm 0.2$ pmol min$^{-1}$ μg$^{-3}$) was much higher than that of Xi'an ($6.9 \pm 3.2$ pmol min$^{-1}$ μg$^{-3}$) (Wang et al., 2020b; Yu et al., 2022c), which was similar to the situation in this study.

In lines 252-253 of the revised manuscript, we have changed "Figure 2 shows the comparison of $DTT_v$ and $DTT_m$ values measured in this study..." to "Figure 2 shows the comparison of water-soluble $PM_{2.5}$ DTT activity measured in this study..."

In lines 254-255 of the revised manuscript, we have changed "...the $DTT_v$ values measured in Beijing (Campbell et al., 2021; Oh et al., 2023; this study)" to "...the $DTT_v$ values measured in Beijing in this study..."

In lines 258-259 of the revised manuscript, we have changed "...higher than that in Xi'an, Nanjing, Hangzhou, Guangzhou, and Shenzhen in China, and Gwangju in Korea..." to "...higher than that in Xi'an, Nanjing, Hangzhou, Guangzhou, and Shenzhen in China..."

PMF analysis. Is n=31 (approximately) sufficient for a robust analysis? Justify.

Response: A 62 (total number of samples collected in the south and north, Beijing) ×
   23 (number of species) matrix was inputted into PMF. The number of samples
   was higher than the number of species, approaching the ratio of at least 3:1
   proposed by Belis et al. (2019). Besides, previous study has reported that if the
   variances among samples are significant, it can obtain physically meaningful
   PMF results (Sun et al., 2011).

In lines 206-209 of the revised manuscript, it now reads "...A total of 62 samples and
   23 species were input into PMF model. The number of samples is higher than the
   number of species, and approaching the ratio of at least 3:1 proposed by Belis et
   al. (2019)."

The correlation analysis is interesting, but a more mechanistic analysis, eg maybe
doing more experiments with the filter samples, as noted above, would add strength to
the conclusions. Example, the sentence on line 363-366 in the Conclusions ("The
results indicate that in the north trace elements and water-soluble organic compounds,
especially BrC…") is a strong statement simply based on correlations.

Response: Thanks for the professional comment. Certainly, it will be interesting and
   helpful to understand the mechanisms of the influence of trace elements and
   organic compounds on the OP of water-soluble $PM_{2.5}$. However, the focus of this
   study was the district differences in water-soluble $PM_{2.5}$ OP and its connection
   with organic compounds, trace elements and sources. The mechanisms of the
   influence of trace elements and organic compounds on the OP of water-soluble
   $PM_{2.5}$ was not the objective of this study. Due to the chemical composition of
   water-soluble $PM_{2.5}$ is very complex, the DTT response of only a few
   compounds has been studied (Charrier and Anastasio, 2012; Xiong et al., 2017),
   and there is no study on the mechanisms by which brown carbon (BrC) affects
   the OP of $PM_{2.5}$. Besides, the interactions between metals and organics, as well
   as between organics and organics, also affect the DTT consumption of
   particulate matter (Yu et al., 2018), making it more difficult to understand their

influence mechanisms. Each of these aspects require intensive studies in the future.

References

Belis, C., Larsen, B. R., Amato, F., Haddad, I. El, Favez, O., Harrison, R. M., Hopke, P. K., Nava, S., Paatero, P., Prévôt, A., Quass, U., Vecchi, R., and Viana, M.: European Guide on Air Pollution Source Apportionment with Receptor Models, JRC References Report, March, 88, 1–170, https://doi.org/10.2788/9307, 2019.

Campbell, S. J., Wolfer, K., Utinger, B., Westwood, J., Zhang, Z. H., Bukowiecki, N., Steimer, S. S., Vu, T. V., Xu, J., Straw, N., Thomson, S., Elzein, A., Sun, Y., Liu, D., Li, L., Fu, P., Lewis, A. C., Harrison, R. M., Bloss, W. J., Loh, M., Miller, M. R., Shi, Z., and Kalberer, M.: Atmospheric conditions and composition that influence $PM_{2.5}$ oxidative potential in Beijing, China, Atmos. Chem. Phys., 21, 5549-5573, 10.5194/acp-21-5549-2021, 2021.

Charrier, J. G. and Anastasio, C.: On dithiothreitol (DTT) as a measure of oxidative potential for ambient particles: evidence for the importance of soluble transition metals, Atmos. Chem. Phys., 12, 9321-9333, 10.5194/acp-12-9321-2012, 2012.

Charrier, J. G., McFall, A. S., Vu, K. K.-T., Baroi, J., Olea, C., Hasson, A., and Anastasio, C.: A Bias in the "Mass-Normalized" DTT Response – An Effect of Non-Linear Concentration Response Curves for Copper and Manganese, Atmos. Environ., 144, 325–334, 2016.

Gao, D., Fang, T., Verma, V., Zeng, L., and Weber, R. J.: A method for measuring total aerosol oxidative potential (OP) with the dithiothreitol (DTT) assay and comparisons between an urban and roadside site of water-soluble and total OP, Atmos. Meas. Tech., 10, 2821-2835, 10.5194/amt-10-2821-2017, 2017.

Oh, S. H., Park, K., Park, M., Song, M., Jang, K. S., Schauer, J. J., Bae, G. N., and Bae, M. S.: Comparison of the sources and oxidative potential of $PM_{2.5}$ during winter time in large cities in China and South Korea, Sci. Total Environ., 859,

160369, 10.1016/j.scitotenv.2022.160369, 2023.

Puthussery, J. V., Dave, J., Shukla, A., Gaddamidi, S., Singh, A., Vats, P., Salana, S., Ganguly, D., Rastogi, N., Tripathi, S. N., and Verma, V.: Effect of Biomass Burning, Diwali Fireworks, and Polluted Fog Events on the Oxidative Potential of Fine Ambient Particulate Matter in Delhi, India, Environ. Sci. Technol., 56, 14605-14616, 10.1021/acs.est.2c02730, 2022.

Sun, Y., Zhang, Q., Zheng, M., Ding, X., Edgerton, E. S., and Wang, X.: Characterization and Source Apportionment of Water-Soluble Organic Matter in Atmospheric Fine Particles ($PM_{2.5}$) with High-Resolution Aerosol Mass Spectrometry and GC–MS, Environ. Sci. Technol., 45, 4854–4861, https://doi.org/10.1021/es200162h, 2011.

Wang, Y., Wang, M., Li, S., Sun, H., Mu, Z., Zhang, L., Li, Y., and Chen, Q.: Study on the oxidation potential of the water-soluble components of ambient $PM_{2.5}$ over Xi'an, China: Pollution levels, source apportionment and transport pathways, Environ. Int., 136, 105515, 10.1016/j.envint.2020.105515, 2020b.

Xiong, Q., Yu, H., Wang, R., Wei, J., and Verma, V.: Rethinking Dithiothreitol-Based Particulate Matter Oxidative Potential: Measuring Dithiothreitol Consumption versus Reactive Oxygen Species Generation, Environ. Sci. Technol., 51, 6507-6514, 10.1021/acs.est.7b01272, 2017.

Yu, H., Wei, J., Cheng, Y., Subedi, K., and Verma, V.: Synergistic and Antagonistic Interactions among the Particulate Matter Components in Generating Reactive Oxygen Species Based on the Dithiothreitol Assay, Environ. Sci. Technol., 52, 2261–2270, 2018.

Yu, Y., Cheng, P., Li, Y., Gu, J., Gong, Y., Han, B., Yang, W., Sun, J., Wu, C., Song, W., and Li, M.: The association of chemical composition particularly the heavy metals with the oxidative potential of ambient $PM_{2.5}$ in a megacity (Guangzhou) of southern China, Environ. Res., 213, 113489, 10.1016/j.envres.2022.113489, 2022c.

Referee #2

This study explored the sources of aerosol oxidative potential – quantified through the DTT assay – at two sites in Beijing. Daily PM$_{2.5}$ filters were collected at both sites for a period of one month. In addition to DTT, water-soluble organics, water-soluble and total metal concentrations, and certain organic markers were measured. PMF was applied in an attempt to apportion the DTT response to different sources. Overall, the manuscript topic is certainly relevant for ACP and some of the results are novel and insightful. However, there are several places where key conclusions are not robustly supported by the data. There are many places where more nuanced analysis is needed.

Specific Comments:

- There are significant limitations with the present study that need to be discussed, and it is definitely not "comprehensive" as the study claims. The limitations are (1) that measurements were only conducted for a period of one month, and (2) water-insoluble species were excluded from the analysis. Therefore, the discussion associated with Fig. 2 needs to be qualified. Similarly, the authors are advised against using descriptors like "exposure-relevant toxicity" and "PM2.5 intrinsic toxicity". Water-insoluble components can contribute to OP (and thus the exposure and toxicity of people in Beijing), yet they were not quantified in this study. The authors should being more accurate/specific with their description of results here and the implications of their findings.

Response: Thanks for your careful reading and professional comments. We have revised the description of "comprehensive". Due to we only collected samples simultaneously at two sites for one month, it is difficult to improve the limitation that the measurements in this study were only conducted for one month. However, in the future, we will consider conducting longer periods of simultaneously sample collection and analysis in different regions.

The studies shown in Figure 2 mainly considered that their sampling time was similar to that of this study, and DTT assay was used. We rechecked the comparative studies in Figure 2 and removed the studies in Beijing, China, and Gwangju,

Korea (Campbell et al., 2021; Oh et al., 2023), which were not extracted samples with water for DTT analysis (Figure R1). Besides, we have revised the descriptions of "exposure-relevant toxicity" and "PM$_{2.5}$ intrinsic toxicity", as well as the descriptions of relevant results to make these expressions more specific.

In lines 99-100 of the revised manuscript, we have changed "...provide a comprehensive comparison of..." to "...provide a comparison of..."

In lines 252-253 of the revised manuscript, we have changed "Figure 2 shows the comparison of DTT$_v$ and DTT$_m$ values measured in this study..." to "Figure 2 shows the comparison of water-soluble PM$_{2.5}$ DTT activity measured in this study..."

In lines 254-255 of the revised manuscript, we have changed "...the DTT$_v$ values measured in Beijing (Campbell et al., 2021; Oh et al., 2023; this study)" to "...the DTT$_v$ values measured in Beijing in this study..."

In lines 258-259 of the revised manuscript, we have changed "...higher than that in Xi'an, Nanjing, Hangzhou, Guangzhou, and Shenzhen in China, and Gwangju in Korea..." to "...higher than that in Xi'an, Nanjing, Hangzhou, Guangzhou, and Shenzhen in China..."

[Figure]

**Figure R1.** Comparison of DTT$_v$ and DTT$_m$ values measured in this study with those measured in other areas of Asia during similar period.

In lines 247-248 of the revised manuscript, we have changed "...indicates that the intrinsic OP of $PM_{2.5}$ was higher in the north than in the south." to "...indicates that the intrinsic OP of water-soluble components of $PM_{2.5}$ was higher in the north than in the south."

In lines 248-250 of the revised manuscript, we have changed "...indicate that the exposure-relevant toxicity of $PM_{2.5}$ was comparable in the two sites" to "...indicate that the exposure-relevant OP of water-soluble components of $PM_{2.5}$ was comparable in the two sites"

In lines 264-266 of the revised manuscript, we have changed "...The differences in $DTT_v$ and $DTT_m$ values in different regions reflect the regional differences in $PM_{2.5}$ exposure risk and intrinsic toxicity, which can be explained by..." to "...The differences in water-soluble DTT activity of $PM_{2.5}$ in different regions can be explained by..."

In lines 407-409 of the revised manuscript, we have changed "...indicating that the $PM_{2.5}$ exposure-relevant toxicity was similar in the two sites and that the $PM_{2.5}$ intrinsic toxicity was higher in the north than in the south." to "...indicating that the exposure-relevant OP of water-soluble components of $PM_{2.5}$ was similar in the two sites and that the intrinsic OP of water-soluble components of $PM_{2.5}$ was higher in the north than in the south."

- In several cases, the authors are cautioned against making the well-known mistake of confusing correlation with causation. Section 3.2 concludes that nitroaromatic compounds "could be important contributors to DTT consumption," on the basis of their correlation with DTTv. To my knowledge, the response of NACs in the DTT assay has not been assessed. Therefore, without this direct knowledge, it might be other components, including those not measured, that correlate with NACs that are driving the correlation. A similar comment applies to discussion surrounding Figure 4, including Line 261-262: "*the consumption of DTT from elements depend primarily on its soluble fraction instead of their total content.*" However, it was only the water-soluble fraction that was added to the

DTT assay, so it makes sense that DTTv would have much stronger correlations with the water-soluble species. The authors are referred to the work of C. Sioutas, who has examined the response of soluble and insoluble PM fractions in the DTT assay.

Response: Thanks for your professional comments. We agree that it might be other components related to NACs driving the correlation between NACs and $DTT_v$.

In line 291 of the revised manuscript, we have changed "...suggesting that NACs could be important contributors..." to "...suggesting that NACs may be important contributors..."

In lines 298-301 of the revised manuscript, it now reads "...Certainly, it may also be other substances related to NACs that contribute to the DTT activity, including those not detected in this study, driving the good correlation between NACs and $DTT_v$ in the north of Beijing, which is worth studying in the future."

In lines 305-306 of the revised manuscript, we have deleted the sentence "suggesting that the consumption of DTT from elements depend primarily on its soluble fraction instead of their total content"

- The PMF results need to be analyzed more critically, and with much more detail. For example, traffic is identified as the most significant contributor to the DTT activity in the south (39.1%). And yet, aside from hopanes, other elements attributed to traffic emissions (Ba, Sr, and Cu) seem to have very low correlation coefficients with DTT? The discussion in Section 3.3 is also confusing: the point of PMF is that it is much more sophisticated than simple linear correlations, yet the discussion here uses correlations with individual species to draw conclusions about the sources contributing to $DTT_v$. Then the discussion moves to the PMF output, but the connection between these is not apparent.

Response: In the south, the correlation coefficients between Ba, Sr, and Cu and $DTT_v$ were indeed low (r < 0.4), however, the concentrations of soluble Ba, Sr, and Cu were higher than most of other trace elements, as described in Section 3.1. The low correlations between these trace elements and $DTT_v$ may be due to the

nonlinear response of DTT activity to their concentrations (Charrier and Anastasic, 2012). Besides, it may be other species (e.g., quinones) from vehicle emissions contribute more to $DTT_v$ than these trace elements, which may also the reason for the weak correlations between these trace element and $DTT_v$. Further, the acquisition of PMF results was much more sophisticated than simple linear correlations, and that the correlation between $DTT_v$ and individual species was not directly related to PMF output. To reduce confusion, the description in Section 3.3 has been revised, and more details about the PMF analysis have been added to the Supporting Information (**PMF analysis**).

In lines 334-357 of the revised manuscript, it now reads, "This study analyzed eight organic markers... to help identify the sources of DTT activity. The correlation coefficients between $DTT_v$ and organic markers are shown in Figure S5...To further quantify the sources of DTT activity in the south and the north of Beijing, the PMF model, which was widely used for the source apportionment of $PM_{2.5}$ OP (Liu et al., 2018; Shen et al., 2022; Cui et al., 2023), was applied. The input species include $DTT_v$, soluble elements and organic markers, and five to seven factors were examined. Due to the oil factor mixed with vehicle emissions factor in the five-factor solution, and there was no new reasonable factor when increasing the factor number to seven in the PMF analysis (Figure S6). Finally, six factors were resolved and quantified using PMF model in the south and north of Beijing..."

In lines 211-212 of the revised manuscript, it now reads "More details are described in SI (PMF analysis)." For more details about PMF analysis, please see SI (**PMF analysis**).

- Many, many method details (blanks, calibration procedures, QA/QC procedures) are missing. The UV-Vis spectrophotometer model is not given. These can be included in the SI, but they are not present at all. Measurement uncertainties were an input into the PMF model (Line 174) yet these were not given for any species, nor the methodology to quantify the uncertainties.

Response: Thanks for pointing these out. All of the results reported in this study were corrected for blanks. The calibration procedures and QA/QC procedures of ICP-MS analysis for trace elements and GC-MS analysis for organic markers were shown in Supplementary Information (**ICP-MS analysis and GC-MS analysis** sections). The quality control of LWCC-UV/Vis analysis for light absorption has been added in Supplementary Information (**Calculation of absorption coefficient of BrC** section). The quality control description of DTT analysis and the model of UV-Vis spectrophotometer has been added in the revised manuscript. The method for quantifying the uncertainty of species input into the PMF model has also been added in Supplementary Information (**PMF analysis**).

In lines 134-135 of the revised manuscript, it now reads "...UV-Vis spectrophotometer (300-700 nm; Ocean Optics, USA)..."

In line 160 of the revised manuscript, it now reads "...All of the results reported in this study were corrected for blanks. "

In lines 181-183 of the revised manuscript, it now reads "...Daily solution blanks and filter blanks were analyzed in parallel with samples to evaluate the consistency of the system performance. Ambient samples were corrected for filter blank."

In lines 210-212 of the revised manuscript, it now reads "...The species-specific uncertainties were calculated following Liu et al. (2017). More details are described in SI (PMF analysis)."

In **Calculation of absorption coefficient of BrC** section in SI, a paragraph has been added, it reads "The light absorption of water-soluble light-absorbing organic compounds (also known as brown carbon, BrC) were measured with an UV-Vis spectrophotometer equipped with a liquid waveguide capillary cell. During the measurement, the system was cleaned with ultrapure water ($> 18.2$ M$\Omega$ cm) after each sample analysis. After cleaning, for instrument calibration, the baseline was zeroed using the Spectra-Suite software so that zero absorption was recorded at all wavelengths for ultrapure water."

The method for quantifying the uncertainty of species input into the PMF model

please see Supplementary Information (**PMF analysis**).

- Also, I acknowledge that the methods the authors have used are widely applied in aerosol studies, however, two potential measurement artifacts need to be acknowledged and discussed. The first relates to potential compositional changes that may occur when the filters are sonicated. Sonication produces hydroxyl radicals and this can change the organic composition (e.g., Miljevic et al., 2014, and references therein). The second potential artifact relates to metal precipitation during the DTT assay. This can cause complex responses in the DTT assay that are not straightforward to interpret (Yalamanchili et al., 2023). Again, the authors have applied established methods here, but these potential effects can (and should) still be discussed.

Response: Thanks for your professional comments. In lines 166-172 of the revised manuscript, it now reads "Several studies have shown that ultrasonic treatment of samples can lead to an increase in its OP values (Miljevic et al., 2014; Jiang et al., 2019), however, there was also a study showed that the difference in OP values of water-soluble $PM_{2.5}$ measured by DTT assay was little for samples extracted by ultrasonic and shaking (Gao et al., 2017). Consistent with the extraction methods of organic markers and trace elements analysis, ultrasonic method was used to extract samples for DTT analysis."

In lines 199-202 of the revised manuscript, it now reads "...This study did not consider the impact of metal precipitation in phosphate matrix on the measured DTT values, as there is no a straightforward method to correct the artifacts caused by this phenomenon (Yalamanchili et al., 2023)."

- I don't follow the explanation in Lines 270-272: why wouldn't the non-linear response also apply in the north?

Response: We agree that the non-linear response of DTT consumption to trace element concentrations was also applicable in the north. DTT consumption has a non-linear response to trace element concentration. With the the increase of trace

element concentration, the increase of DTT consumption rate decreased, therefore, the linear correlation between trace elements and DTT activity was higher in the low concentration range than in the high concentration range (Charrier and Anastasio, 2012). Because the concentration of soluble elements in the north (total of $99.2 \pm 83.4$ ng m$^{-3}$) was much lower than that in the south (total of $185.4 \pm 116.7$ ng m$^{-3}$), the non-linear response of DTT consumption to trace element concentrations could have greater impact on the correlations between DTT activity and soluble elements in the south than in the north.

**Technical Corrections:**

- Should all WSOC units be μg-C m-3 (instead of μg m-3)?

Response: Thanks for pointing this out. The unit of WSOC has been changed from μg m$^{-3}$ to μgC m$^{-3}$.

- Does Fig. S3 and Table S1 together indicate that only ~1-2% of Fe was soluble?

Response: Yes. The average Fe solubility in the south and north of Beijing was $1.8 \pm 1.2\%$ and $1.2 \pm 1.0\%$, respectively, which was similar to the value in Qingdao, China ($1.3 \pm 1.4\%$) (Zhang et al., 2022).

- Figure S1 needs a scale so the distance between the sites can be estimated.

Response: Thanks for pointing this out. The distance between the two sites is about 42 km and a scale bar has been added to Figure S1 (Figure R2).

In lines 110-111 of the revised manuscript, it now reads "...The distance between the two sampling sites is about 42 km."

[Figure]

**Figure R2.** Map of the sampling sites. NCNT and DFZ are abbreviations for the north (the National Center for Nanoscience and Technology) and south (the Dingfuzhuang village, Daxing district) sites of Beijing, China, respectively. The left panel from Ministry of Natural Resources of China, and the right panel from Google Maps.

- Line 17-18: sentence needs clarification

Response: Thank you. In lines 17-19 of the revised manuscript, we have changed "...atmospheric fine particles, while our understanding of their relationship is still limited." to "...atmospheric fine particles ($PM_{2.5}$), while our understanding of water-soluble $PM_{2.5}$ OP and its sources, as well as its relationship with water-soluble components, is still limited."

- Line 65: "organic" should be "organics"

Response: Thank you. Change made.

- Line 116: "foils" should be "foil"

Response: Because one sample was wrapped in an aluminum foil, there were multiple samples, so "foils" was used here.

- Line 134: change "were" to "was"

Response: Thank you. Change made.

- Line 141: edit sentence for clarity

Response: Thank you. In lines 144-146 of the revised manuscript, we have changed "...another filter with same size was used and digestion after added of 10 mL $HNO_3$ and 1 mL HF." to "...another 47 mm diameter filter of the same sample was used and digested with 10 mL $HNO_3$ and 1 mL HF at 180 ℃ for 12 h."

- Line 207: edit sentence for grammar

Response: Thank you. In lines 245-246 of the revised manuscript, we have changed "...may be due to that the increased $PM_{2.5}$ in the south contains more substances..." to "...may be due to the increased $PM_{2.5}$ in the south containing more substances...."

- Line 244: suggest changing "are coincide" to "qualitatively agree" or similar

Response: Thank you. In line 284 of the revised manuscript, we have changed "...are coincide..." to "...qualitatively agree..."

- Line 314: change "wither" to "winter"

Response: Thank you. Change made.

- Paragraph beginning on Line 337: is it accurate to qualify these as "regional" differences?

Response: Thank you. We have changed "regional" to "district".

In line 390 of the revised manuscript, we have changed "...exhibiting obvious regional differences." to "...exhibiting obvious district differences."

In line 395-396 of the revised manuscript, we have changed "The large regional differences in sources of $DTT_v$... " to "The large district differences in sources of $DTT_v$..."

**References**

Miljevic, B., et al., To Sonicate or Not to Sonicate PM Filters: Reactive Oxygen Species Generation Upon Ultrasonic Irradiation, *Aerosol Science and Technology*, 48: 1276-1284, 2014.

Yalamanchili, J., et al., Measurement artifacts in the dithiothreitol (DTT) oxidative potential assay caused by interactions between aqueous metals and phosphate buffer, *Journal of Hazardous Materials*, 465, 131693, 2023.

**References**

Campbell, S. J., Wolfer, K., Utinger, B., Westwood, J., Zhang, Z. H., Bukowiecki, N., Steimer, S. S., Vu, T. V., Xu, J., Straw, N., Thomson, S., Elzein, A., Sun, Y., Liu, D., Li, L., Fu, P., Lewis, A. C., Harrison, R. M., Bloss, W. J., Loh, M., Miller, M. R., Shi, Z., and Kalberer, M.: Atmospheric conditions and composition that influence $PM_{2.5}$ oxidative potential in Beijing, China, Atmos. Chem. Phys., 21, 5549-5573, 10.5194/acp-21-5549-2021, 2021.

Charrier, J. G. and Anastasio, C.: On dithiothreitol (DTT) as a measure of oxidative potential for ambient particles: evidence for the importance of soluble transition metals, Atmos. Chem. Phys., 12, 9321-9333, 10.5194/acp-12-9321-2012, 2012.

Cui, Y., Zhu, L., Wang, H., Zhao, Z., Ma, S., and Ye, Z.: Characteristics and Oxidative Potential of Ambient $PM_{2.5}$ in the Yangtze River Delta Region: Pollution Level and Source Apportionment, Atmosphere, 14, 10.3390/atmos14030425, 2023.

Gao, D., Fang, T., Verma, V., Zeng, L., and Weber, R. J.: A method for measuring total aerosol oxidative potential (OP) with the dithiothreitol (DTT) assay and comparisons between an urban and roadside site of water-soluble and total OP, Atmos. Meas. Tech., 10, 2821-2835, 10.5194/amt-10-2821-2017, 2017.

Jiang, H., Xie, Y., Ge, Y., He, H., and Liu, Y.: Effects of ultrasonic treatment on dithiothreitol (DTT) assay measurements for carbon materials, J. Environ. Sci., 84, 51–58, 2019.

Liu, W., Xu, Y., Liu, W., Liu, Q., Yu, S., Liu, Y., Wang, X., and Tao, S.: Oxidative potential of ambient $PM_{2.5}$ in the coastal cities of the Bohai Sea, northern

China: Seasonal variation and source apportionment, Environ. Pollut., 236, 514-528, 10.1016/j.envpol.2018.01.116, 2018.

Liu, Y., Yan, C. Q., Ding, X., Wang, X. M., Fu, Q. Y., Zhao, Q. B., Zhang, Y. H., Duan, Y. S., Qiu, X. H., and Zheng, M.: Sources and spatial distribution of particulate polycyclic aromatic hydrocarbons in Shanghai, China, Sci. Total Environ., 584-585, 307-317, https://doi.org/10.1016/j.scitotenv.2016.12.134, 2017.

Miljevic, B., Hedayat, F., Stevanovic, S., Fairfull-Smith, K. E., Bottle, S. E., and Ristovski, Z. D.: To sonicate or not to sonicate PM filters: reactive oxygen species generation upon ultrasonic irradiation, Aerosol. Sci. Technol., 48, 1276–1284, 2014.

Oh, S. H., Park, K., Park, M., Song, M., Jang, K. S., Schauer, J. J., Bae, G. N., and Bae, M. S.: Comparison of the sources and oxidative potential of $PM_{2.5}$ during winter time in large cities in China and South Korea, Sci. Total Environ., 859, 160369, 10.1016/j.scitotenv.2022.160369, 2023.

Shen, J., Taghvaee, S., La, C., Oroumiyeh, F., Liu, J., Jerrett, M., Weichenthal, S., Del Rosario, I., Shafer, M. M., Ritz, B., Zhu, Y., and Paulson, S. E.: Aerosol Oxidative Potential in the Greater Los Angeles Area: Source Apportionment and Associations with Socioeconomic Position, Environ. Sci. Technol., 56, 17795-17804, 10.1021/acs.est.2c02788, 2022.

Yalamanchili, J., Hennigan, C. J., and Reed, B. E.: Measurement artifacts in the dithiothreitol (DTT) oxidative potential assay caused by interactions between aqueous metals and phosphate buffer, J. Hazard. Mater., 456, 131693, 2023.

Zhang, H., Li, R., Dong, S., Wang, F., Zhu, Y., Meng, H., Huang, C., Ren, Y., Wang, X., Hu, X., Li, T., Peng, C., Zhang, G., Xue, L., Wang, X., and Tang, M.: Abundance and fractional solubility of aerosol iron during winter at a coastal city in northern China: Similarities and contrasts between fine and coarse particles, J. Geophys. Res.-Atmos., 127, e2021JD036070, 2022.

---

## Author Response (AR2)

The authors thank the referees to review our manuscript and particularly for the valuable comments and suggestions that have significantly improved the manuscript. We provide below point-by-point responses (in blue) to the referees' comments and have made changes accordingly in the revised manuscript.

Referee #1

I thank the authors for addressing the comments from the first review. I have two issues I wish the authors consider.

Overall, I am still concerned with the use of variable concentrations in the extraction vial that may be affecting the results and leading to erroneous conclusions. The authors response based on plotting $DTT_m$ vs $PM_{2.5}$ mass and stating that at high $PM_{2.5}$ mass concentrations $DTT_m$ does not depend on mass is not highly convincing, in part because one is dividing by a large number ($PM_{2.5}$ mass). The authors ideally should re-evaluate some of the filters (if some fraction of the filters remain). Example, redo the DTT analysis for say three to four different concentrations of PM in the reaction vial including the concentration recommended in a number of publications and see if this affects $DTT_m$. If there are no filters to redo some further analysis, then state that. Overall, this issue is noted in the manuscript so readers can assess for themselves if this is a significant limitation with this study or not. I find the new added explanation given rather weak, but I do not feel it should hold up the publication of this work since it has been noted.

Response: Thank you for your professional comments and valuable suggestions. Figure S2 shows the effect of $PM_{2.5}$ concentration in the reaction vial on $DTT_m$ ($PM_{2.5}$ concentration in the reaction vial changed from 2 to 300 μg mL$^{-1}$, including the concentration recommended in previous publications) (Figure R1). The concentration of $PM_{2.5}$ in the reaction vial does indeed have an impact on $DTT_m$. In the range of $PM_{2.5}$ concentration in the reaction vial less than 150 μg mL$^{-1}$, the $DTT_m$ response decreased significantly with the increase of $PM_{2.5}$ concentration in the reaction vial. However, when the concentration of $PM_{2.5}$ in

the reaction vial is greater than 150 µg mL$^{-1}$, the DTT$_m$ response changes little (< 12%) with the increase of PM$_{2.5}$ concentration in the reaction vial. Because most samples did not have enough filters to redo further analysis, therefore, not every sample was analyzed for the effect of PM$_{2.5}$ concentration in the reaction vial on DTT$_m$ response. However, in this study, most of samples (> 80%) had a concentration of PM$_{2.5}$ in the reaction vial greater than 150 µg mL$^{-1}$, therefore, the difference in PM$_{2.5}$ concentration in reaction vial of different samples should had a relatively small impact on the difference in DTT$_m$ values of different samples.

To make the expression clearer, in lines 186-204 of the revised manuscript, it now reads, "Considering that for samples containing a significant amount of substances whose DTT response is non-linear with PM$_{2.5}$ concentration (e.g., Cu, Mn), the DTT$_m$ value depends on the concentration of PM$_{2.5}$ added to the reaction solution... The response of DTT$_m$ to PM$_{2.5}$ concentration added to the reaction solution was analyzed using sample containing high concentrations of soluble Cu and Mn (Figure S2). When the PM$_{2.5}$ concentration added to the reaction solution is less than 150 µg mL$^{-1}$, the DTT$_m$ response is greatly affected by the difference in added PM$_{2.5}$ concentration; however, when the PM$_{2.5}$ concentration added to the reaction solution is greater than 150 µg mL$^{-1}$, the DTT$_m$ response is less affected by the difference in PM$_{2.5}$ concentration (< 12%). In this study, the concentration of PM$_{2.5}$ added to the reaction solution of most samples from the two sites was greater than 150 µg mL$^{-1}$ (ranged from 78.7 to 748.7 µg mL$^{-1}$, with an average of 408.9 ± 164.1 and 206.6 ± 95.0 µg mL$^{-1}$ in the south and north, respectively), therefore, the difference in PM$_{2.5}$ concentration added to the reaction solution of different samples should had a relatively small impact on the difference in DTT$_m$ values of different samples."

[Figure]

**Figure R1.** Measured DTT$_m$ response as a function of PM$_{2.5}$ concentration in the reaction vial.

One further clarification is requested. In response to a question the authors have added: "...A total of 62 samples and 23 species were input into PMF model. The number of samples is higher than the number of species, and approaching the ratio of at least 3:1 proposed by Belis et al. (2019)."

I suggest writing as. A PMF analysis was performed for each site based on 31 filter samples collected at each site, and 23 species were input into each of the PMF models. The number of samples is higher than the number of species." I believe the latter part is wrong ("and approaching the ratio of at least 3:1") since the ratio should be 31/23 not 62/23 which does not approach 3:1.

Response: Thanks for your suggestion. In lines 210-213 of the revised manuscript, it now reads, "...For each site, 31 samples and 23 species were input into PMF model. The number of samples is higher than the number of species."

Referee #3

In this manuscript, Yuan et al., report upon an analysis of $PM_{2.5}$ composition and oxidative potential (OP: measured using the DTT assay), for two locations in the Beijing area. The compositional analysis is pretty extensive, including both metals and some organic species. They link their composition analysis with PMF to assess source impacts on OP.

As a measurement report, this paper achieves its goal, i.e., contributing to the databank of PM composition and OP characterization. The further analyses, including source apportionment and such, are similar to past work, and only go a little ways in to what might be done with their data. Their findings are similar to prior work as well. A major shortcoming is they tend not to answer the big question of why does OP vary so much between north and south. DTTm is rather different between north and south, but the sources contributing to PM are rather similar (Fig. 5), one has to ask "Why?" That question is not really addressed. Why are the correlations between DTT OP and soluble elements so different as well? Is it really due to the non-linear response as a function of level? That seems a bit far-fetched as the levels are not that different, and unless that non-linearity becomes very large (which would tend to suggest criticism of any further analysis or utility of the measurements), it would not be expected to have such a large impact on the correlations. If you plot metal concentrations vs. OP, does a non-linearity appear? It should also be noted that the correlation between DTTv and WSOC and ABS is also much less in the south: might there be something more about the measurements in the south? Given what is currently there, one might suggest some further caveats and cautions about what might be taken away from the analyses.

Response: Thanks for your professional comments. In this study, the difference of OP between the south and north of Beijing were mainly ascribed to the differences in chemical composition and sources, as well as the interactions between metals and organic compounds. We analyzed the differences in trace elements and WSOC between the south and north, which are substances that previous studies have shown to contribute significantly to DTT activity. In lines 237-240 of the revised manuscript, it reads "...These results indicate that the sources and

emission strength of water-soluble organic compounds were different in the south and north of Beijing, suggesting the different contribution of water-soluble organic compounds to DTT activity." In lines 249-251 of the revised manuscript, it reads "...The lower $DTT_m$ in the south than in the north may be due to the increased $PM_{2.5}$ in the south containing more substances with no or little contribution to DTT activity..." The sources of $DTT_v$ in the south and north were quantified using the PMF model. The relative contribution of each source to $DTT_v$ may not be very different in the south and the north, but their absolute contribution difference is 1.2-3.4 times (Table R1). In lines 393-395 of the revised manuscript, it now reads "…The absolute contribution of each source to $DTT_v$ varies by 1.2-3.4 times between the south and north of Beijing (Table S2)." Due to the complex chemical composition of water-soluble $PM_{2.5}$, and the complex effect of interactions between metals and organics, as well as between organics and organics on DTT consumption of PM (Yu et al., 2018), it is difficult to understand their influence mechanisms. Each of these aspects require intensive studies in the future. Therefore, this study only mentioned that the interactions between metals and organic compounds can also affect DTT activity. In lines 319-322 of the revised manuscript, it reads "...the interactions between metals and organic compounds also affect the consumption of DTT..., with both synergistic and antagonistic effects. For example..."

Figure R1 shows the relationship between soluble trace elements and $DTT_v$. Generally, the relationship between most soluble trace elements and $DTT_v$ was more non-linear than linear. Besides, the concentration of soluble trace elements was generally higher in the south than in the north (1.3-4.1 times) (Figure R2). These results affect the differences in correlation between DTT OP and soluble elements in the south and north. In addition, the interactions between metals and organic compounds also play a role, as described above. In lines 315-318 of the revised manuscript, it now reads "…As shown in Figure S5, the relationship between most soluble trace elements and $DTT_v$ was more non-linear than linear. As the concentration of soluble elements increases, the growth rate of $DTT_v$

obviously decreases."

The differences in correlation between $DTT_v$ and WSOC and Abs in the south and north were mainly ascribed to the differences in chemical composition and sources of water-soluble PM. Of course, interactions between organics and between organics and metals could also have an impact (Yu et al., 2018). In this study, we analyzed the differences in the concentration of nitroaromatic compounds (NACs) and their correlation with $DTT_v$ between the south and north. Due to the complex chemical composition of water-soluble organic matter, it is difficult to investigate the differences in the effects of interactions between organics and between organics and metals on DTT activity between the south and north. Therefore, there were no more measurements in the south. The analysis of this study suggests that the water-soluble $PM_{2.5}$ OP is closely related to its chemical composition and sources, and the effect of interactions between organics and between metals and organics on $PM_{2.5}$ OP is still worthy of further study. In lines 327-330 of the revised manuscript, it reads "...Due to the complex composition of water-soluble organic aerosols, the knowledge about the effects of organics and metal-organic interactions on DTT activity are still limited, especially the effects of BrC chromophores and their interactions with metals." In lines 395-397 of the revised manuscript, it reads "...The large district differences in sources of $DTT_v$ of water-soluble $PM_{2.5}$ call for more research on the relationship between sources, chemical composition, formation processes and OP of $PM_{2.5}$."

Table R1. DTTv values in the south and north of Beijing and the sources contributions.

|  | South | North |
|---|---|---|
| $DTT_v$ (nmol min$^{-1}$ m$^{-3}$) | 3.9 | 3.5 |
| | | |
| Sources contribution to $DTT_v$ (%) | | |
| Biomass burning | 25.2 | 8.4 |
| Coal burning | 15 | 19.9 |
| Traffic-related | 39.1 | 51.6 |
| Secondary formation | 17.2 | 13 |
| Dust | 2 | 3 |
| Oil combustion | 1.5 | 4.1 |
| | | |
| Sources contribution to $DTT_v$ (nmol min$^{-1}$ m$^{-3}$) | | |
| Biomass burning | 0.98 | 0.29 |
| Coal burning | 0.59 | 0.7 |
| Traffic-related | 1.5 | 1.8 |
| Secondary formation | 0.67 | 0.46 |
| Dust | 0.08 | 0.11 |
| Oil combustion | 0.06 | 0.14 |

[Figure]

Figure R1. Relationship between soluble trace elements concentration (ng m$^{-3}$) and DTT$_v$ (nmol min$^{-1}$ m$^{-3}$).

[Figure]

**Figure R2.** Concentrations of soluble elements in the south and north of Beijing.

The authors have adequately responded to the prior review. I will add, that further uncertainty analysis is suggested, and they should watch the precision with which they report their values. In particular, what is their reproducibility of sampled OP and how was reproducibility tested? Reproducibility of DTT-based OP assays is an issue: not that the measurements are "wrong", just that they can vary by day of analysis given handling, dilution, extraction and other issues. Further, there is the issue of non-linear response brought up. Given the variability in their results and uncertainties in the methods, two significant figures are more than enough.

Response: Thanks for your careful reading and professional comments. The uncertainty analysis of organic compounds and trace elements are shown in Supporting Information (ICP-MS analysis and GC-MS analysis). For DTT analysis, for every 10 samples, select one sample to measure three times to check the reproducibility, and the relative standard deviation was lower than 5%. In lines 180-182 of the revised manuscript, it now reads "...Besides, for every 10 samples, one sample was chosen to be measured three times to check the reproducibility, and the relative standard deviation was lower than 5%." Further, in the revised manuscript, all data reported in this study have been changed to two significant figures.

Finally, the authors should have the manuscript grammar checked again, particularly the sections modified or added in response to the last reviews. While generally the article is reasonably good grammatically, there were some sections (e.g., the paragraph starting "Consider that for samples…") that were in less good shape.

Response: Thank you for your suggestion. We have re-checked the grammar of the manuscript and made corresponding modifications. The paragraph starting with "Consider that for samples" now reads "Considering that for samples containing a significant amount of substances whose DTT response is non-linear with $PM_{2.5}$ concentration (e.g., Cu, Mn), the $DTT_m$ value depends on the concentration of $PM_{2.5}$ added to the reaction solution (Charrier et al., 2016). The response of $DTT_m$ to $PM_{2.5}$ concentration added to the reaction solution was analyzed using sample containing high concentrations of soluble Cu and Mn (Figure S2). When the $PM_{2.5}$ concentration added to the reaction solution is less than 150 µg mL$^{-1}$, the $DTT_m$ response is greatly affected by the difference in added $PM_{2.5}$ concentration; however, when the $PM_{2.5}$ concentration added to the reaction solution is greater than 150 µg mL$^{-1}$, the $DTT_m$ response is less affected by the difference in $PM_{2.5}$ concentration (< 12%). In this study, the concentration of $PM_{2.5}$ added to the reaction solution of most samples from the two sites was greater than 150 µg mL$^{-1}$ (ranged from 79 to 749 µg mL$^{-1}$, with an average of 409 ± 164 and 207 ± 95 µg mL$^{-1}$ in the south and north, respectively), therefore, the difference in $PM_{2.5}$ concentration added to the reaction solution of different samples should had a relatively small impact on the difference in $DTT_m$ values of different samples. This study did not consider the impact of metal precipitation in phosphate matrix on the measured DTT values, as there is not a straightforward method to correct the artifacts caused by this phenomenon (Yalamanchili et al., 2023)." For other changes, please see the revised manuscript with modification marks.

References

Charrier, J. G., McFall, A. S., Vu, K. K.-T., Baroi, J., Olea, C., Hasson, A., and Anastasio, C.: A Bias in the "Mass-Normalized" DTT Response – An Effect of Non-Linear Concentration Response Curves for Copper and Manganese, Atmos. Environ., 144, 325–334, 2016.

Yalamanchili, J., Hennigan, C. J., and Reed, B. E.: Measurement artifacts in the dithiothreitol (DTT) oxidative potential assay caused by interactions between aqueous metals and phosphate buffer, J. Hazard. Mater., 456, 131693, 2023.

Yu, H., Wei, J., Cheng, Y., Subedi, K., and Verma, V.: Synergistic and Antagonistic Interactions among the Particulate Matter Components in Generating Reactive Oxygen Species Based on the Dithiothreitol Assay, Environ. Sci. Technol., 52, 2261–2270, 2018.

---

## Author Response (AR3)

The authors thank the referees to review our manuscript and particularly for the valuable comments and suggestions that have significantly improved the manuscript. We provide below point-by-point responses (in blue) to the referees' comments and have made changes accordingly in the revised manuscript.

Referee #1

The authors have addressed all my questions. Thanks.

Response: Thank you for reviewing our manuscript and making valuable comments and suggestions, which have greatly improved the manuscript. Thank you very much.

Referee #4

The manuscript has some values as a measurement report because it adds a dataset for the characterization of oxidative potential in conditions of high concentrations with a good chemical analysis including also insoluble and soluble trace elements that may be useful to the scientific community. The post processing with correlations and application of receptor model is quite standard with limited novelty. Nevertheless the publication of the manuscript may be valuable. I understand that is already a revised version, nonetheless, there are some aspects not clear, see my specific comments, that should be addressed in a further revision step before considering the paper for publication.

Specific comments

I found a little confusing that the same symbols are used for total and soluble trace elements. Authors should clearly mention in caption and in text to what fraction they refer when discussing correlation, concentrations and so on.

Response: Thanks. We have re-checked the manuscript and made corresponding modifications to any confusion. In line 245 of the revised manuscript, we changed "...the concentration trends of trace elements" to "...the concentration

trends of total trace elements" In line 807 of the revised manuscript, we changed "...(e) elements." to "...(e) total elements." The elements in table S1 have also been specified as total elements.

Line 43. I would not say one of the main but one possible mechanism.

Response: Thanks. Change made.

Lines 43-46. I would suggest to mention the recent work of Guascito et al (Journal of Hazardous Materials 448, 130872, 2023) that correlates oxidative potential with biological effects at different sites.

Response: Thank you for your suggestion and this work is worth mentioning. This work has been added as a reference. In line 45 of the revised manuscript, it now reads "... Guascito et al., 2023)."

Lines 83-98. In this part discussing the literature findings, I suggest to mention the efforts in studying size segregated oxidative potential, for example Besis et al (Toxics 11 (1), 59, 2023).

Response: Thank you for your suggestion. In lines 90-98 of the revised manuscript, it now reads, "...Some studies have also measured the OP of particles with different particle sizes, and reported that smaller size fractions typically have higher ROS activity compared to large PM size fractions (Saffari et al., 2014; Shafer et al., 2016; Besis et al., 2023). For example, Besis et al. (2023) measured the OP of water-soluble fraction of size segregated PM (< 0.49, 0.49-0.95, 0.95-1.5, 1.5-3.0, 3.0-7.2 and > 7.2 µm) collected during the cold and warm periods at an urban site in Thessaloniki, northern Greece, and the results showed that the total DTT activity of the PM < 3 µm size fraction were higher (2-5 times) than that of PM > 3 µm size fraction in both warm and cold periods."

Line 167. Better small rather than little.

Response: Thanks. Change made.

It is mentioned the analysis of levogucosan, mannosan and others. These are used in PMF but not reported in the tables, why? I strongly suggest to have a table with all measured concentrations.

Response: Thank you for pointing this out. The concentrations of measured organic markers (including levoglucosan, mannosan, galactosan and hopanes) have now been added to Table S2. In lines 346-347 of the revised manuscript, it now reads, "...The average concentrations of these organic markers are shown in Table S2."

PMF application. It is not clear if two separate runs have been done of only one. Because in some parts they speak about a limited number of samples like if two runs were done. However, results only include one series of profiles like if all samples were pooled together in a single run. This must be explained. I believe that, considering the limited number of samples, the option to use a single run with all samples may give more stable results from a statistical point of view. In any case, it is necessary to give more details on the PMF results and approach. For example, weak or bad chemical species; why it has not been used the WSOC? Stability of the solution in terms of rotational ambiguity, bootstrap and so on. I suggest to have a look at the work of Belis et al, (Atmospheric environment: X 5, 100053, 2020) for recpeotr models preformances.

Response: Thank you for your careful reading and professional comments. In this study, all samples were input into one single run, so there was only one series of profiles. In lines 214-215 of the revised manuscript, it now reads "...For each site, 31 samples (a total of 62 samples) and 23 species were input into PMF model."

In this study, the source apportionment of $DTT_v$ was performed using PMF as implemented by the multilinear engine (ME-2; Paatero, 1997) via the source-finder (SoFi) interface written in Igor WaveMetrics (Canonaco et al., 2013). Compared to EPA PMF, which does not require any prior information but may have a substantial degree of rotational ambiguity (Paatero, 1997), ME-2 can partly constrain the factors based on a priori information (such as factor profiles)

to reduce rotational ambiguity and direct the solution towards environmentally-meaningful rotations (Huang et al., 2014; Lin et al., 2018). In PMF&ME-2 model, species cannot be set as "Strong", "Weak" or "Bad". The species input into the PMF model all have concentrations much higher than the corresponding uncertainty values, with a signal-to-noise ratio (S/N) > 2. In line 79 of the Supporting Information, it now reads "All data input into PMF model has a signal-to-noise ratio (S/N) greater than 2."

In order to obtain more accurate and refined source apportionment results, only tracer species were input into PMF model, and thus WSOC and NACs were not used.

In PMF&ME-2 model, there are no bootstrap (BS), displacement (DISP), and bootstrap combined with displacement (BS-DISP) results. To reduce rotational ambiguity and obtain an environmentally reasonable solution, the factor profiles were partly constrained in PMF&ME-2 model according to previous studies (Huang et al., 2014; Wang et al., 2017), as shown in Table R1 (added as Table S1). The uncertainties for PMF analysis of these sources were 2-14%. In lines 219-220 of the revised manuscript, it now reads, "...For a clear separation of sources profiles, the contribution of corresponding markers was set to 0 in the sources unrelated to the markers (see Table S1)." In lines 368-369 of the revised manuscript, it now reads, "...The uncertainties of PMF analysis for these sources were 2-14%."

**Table R1.** *F* matrix elements constrained in the ME-2/chemical species 6 factors solution. The 0 value denote the $f_{h,j}$ values constrained in ME-2, while hyphens denote unconstrained elements.

| Species | Biomass burning | Coal Burning | Traffic-related | Secondary Formation | Dust | Oil combustion |
|---|---|---|---|---|---|---|
| DTTv | - | - | - | - | - | - |
| Ti | - | - | - | 0 | - | - |
| V | - | - | - | 0 | - | - |
| Cr | - | - | - | 0 | - | - |
| Mn | - | - | - | 0 | - | - |
| Fe | - | - | - | 0 | - | - |
| Co | - | - | - | 0 | - | - |
| Ni | - | - | - | 0 | - | - |
| Cu | - | - | - | 0 | - | - |
| Zn | - | - | - | 0 | - | - |
| As | - | - | - | 0 | - | - |
| Sr | - | - | - | 0 | - | - |
| Cd | - | - | - | 0 | - | - |
| Ba | - | - | - | 0 | - | - |
| Pb | - | - | - | 0 | - | - |
| Picene | - | - | - | 0 | 0 | - |
| Hopanes | 0 | - | - | 0 | 0 | 0 |
| Galactosan | - | 0 | 0 | 0 | 0 | 0 |
| Mannosan | - | 0 | 0 | 0 | 0 | 0 |
| Levoglucosan | - | 0 | 0 | 0 | 0 | 0 |
| *o*-ph | 0 | 0 | 0 | - | 0 | 0 |
| *m*-ph | 0 | 0 | 0 | - | 0 | 0 |
| *p*-ph | 0 | 0 | 0 | - | 0 | 0 |

The discussion of the difference between results in North and South is not very conclusive, probably further studies with longer datasets are needed and this should be reported in the conclusions (updating lines 421-424). In addition, in lines 240-255, I would suggest to mention that there are other studies showing that samples with similar concentrations may have a strong differences in DTT activity or the other way around interpreted by difference in chemical composition as well as possible antagonistic and synergistic effects, see for example, Lionetto et al (Atmosphere 12 (4), 464, 2021).

Response: Thank you for your professional comments and suggestions. In lines

436-439 of the revised manuscript, it now reads "...Besides, in order to gain a more comprehensive understanding of the regional differences in PM$_{2.5}$ OP, sources and its relationship with chemical composition, longer periods and different seasonal datasets are also need to be studied in the future." In lines 253-257 of the revised manuscript, it now reads "... Ahmad et al. (2021) also reported that the concentrations of PM$_{2.5}$, WSOC, and most elements in Lahore, Pakistan, were higher than those in Peshawar, Pakistan, while the DTT$_v$ values of the two sites were similar, and the DTT$_m$ value in Peshawar was higher than that in Lahore." In lines 263-266 of the revised manuscript, it now reads "...Due to the complex chemical composition of PM$_{2.5}$, there may also be antagonistic and synergistic effects, contributing to the inconsistent relationship between DTT activity and compounds content(Xiong et al., 2017; Lionetto et al., 2021)."

Line 281. Not toxicity here just oxidative potential.

Response: Thanks. Change made.

Figure 1. Are these total or soluble data for trace elements? The same for Table S1.

Response: Figure 1 and Table S1 both show total trace elements. Corresponding clarifications have been made.

Figure 3. Just use a single threshold for statistical significance. It is not necessary to use two. The same for fig. S6.

Response: Change made.

**References**

Ahmad, M., Yu, Q., Chen, J., Cheng, S., Qin, W., and Zhang, Y.: Chemical characteristics, oxidative potential, and sources of PM2.5 in wintertime in Lahore and Peshawar, Pakistan, J. Environ. Sci., 102, 148-158, 10.1016/j.jes.2020.09.014, 2021.

Besis, A., Romano, M. P., Serafeim, E., Avgenikou, A., Kouras, A., Lionetto, M. G., Guascito, M. R., De Bartolomeo, A. R., Giordano, M. E., Mangone, A., Contini, D., and Samara, C.:

Size-Resolved Redox Activity and Cytotoxicity of Water-Soluble Urban Atmospheric Particulate Matter: Assessing Contributions from Chemical Components, Toxics, 11, 10.3390/toxics11010059, 2023.

Canonaco, F., Crippa, M., Slowik, J. G., Baltensperger, U., and Prévôt, A. S. H.: SoFi, an IGOR-based interface for the efficient use of the generalized multilinear engine (ME-2) for the source apportionment: ME-2 application to aerosol mass spectrometer data, Atmos. Meas. Tech., 6, 3649-3661, 10.5194/amt-6-3649-2013, 2013.

Guascito, M. R., Lionetto, M. G., Mazzotta, F., Conte, M., Giordano, M. E., Caricato, R., De Bartolomeo, A. R., Dinoi, A., Cesari, D., Merico, E., Mazzotta, L., and Contini, D.: Characterisation of the correlations between oxidative potential and in vitro biological effects of $PM_{10}$ at three sites in the central Mediterranean, J. Hazard. Mater., 448, 130872, 10.1016/j.jhazmat.2023.130872, 2023.

Huang, R. J., Zhang, Y., Bozzetti, C., Ho, K. F., Cao, J. J., Han, Y., Daellenbach, K. R., Slowik, J. G., Platt, S. M., Canonaco, F., Zotter, P., Wolf, R., Pieber, S. M., Bruns, E. A., Crippa, M., Ciarelli, G., Piazzalunga, A., Schwikowski, M., Abbaszade, G., Schnelle-Kreis, J., Zimmermann, R., An, Z., Szidat, S., Baltensperger, U., El Haddad, I., and Prevot, A. S.: High secondary aerosol contribution to particulate pollution during haze events in China, Nature, 514, 218-222, 10.1038/nature13774, 2014.

Lin, C., Huang, R.-J., Ceburnis, D., Buckley, P., Preissler, J., Wenger, J., Rinaldi, M., Facchini, M. C., O'Dowd, C., and Ovadnevaite, J.: Extreme air pollution from residential solid fuel burning, Nat. Sustain., 1, 512-517, 10.1038/s41893-018-0125-x, 2018.

Lionetto, M., Guascito, M., Giordano, M., Caricato, R., De Bartolomeo, A., Romano, M., Conte, M., Dinoi, A., and Contini, D.: Oxidative Potential, Cytotoxicity, and Intracellular Oxidative Stress Generating Capacity of $PM_{10}$: A Case Study in South of Italy, Atmosphere, 12, 10.3390/atmos12040464, 2021.

Paatero, P.: Least squares formation of robust non-negative factor analysis, Chemom. Intell. Lab., 37, 23-35, https://doi.org/10.1016/S0169-7439(96)00044-5, 1997.

Saffari, A., Daher, N., Shafer, M. M., Schauer, J. J., and Sioutas, C.: Global perspective on the oxidative potential of airborne particulate matter: a synthesis of research findings, Environ. Sci. Technol., 48, 7576-7583, 10.1021/es500937x, 2014.

Shafer, M. M., Hemming, J. D., Antkiewicz, D. S., and Schauer, J. J.: Oxidative potential of size-fractionated atmospheric aerosol in urban and rural sites across Europe, Faraday Discuss., 189, 381-405, 10.1039/c5fd00196j, 2016.

Wang, Q., He, X., Huang, X. H. H., Griffith, S. M., Feng, Y., Zhang, T., Zhang, Q., Wu, D., and Yu, J. Z.: Impact of Secondary Organic Aerosol Tracers on Tracer-Based Source Apportionment of Organic Carbon and PM2.5: A Case Study in the Pearl River Delta, China, ACS Earth Space Chem., 1, 562-571, 10.1021/acsearthspacechem.7b00088, 2017.

Xiong, Q., Yu, H., Wang, R., Wei, J., and Verma, V.: Rethinking Dithiothreitol-Based Particulate Matter Oxidative Potential: Measuring Dithiothreitol Consumption versus Reactive Oxygen Species Generation, Environ. Sci. Technol., 51, 6507-6514, 10.1021/acs.est.7b01272, 2017.